# Cortical oscillations support sampling-based computations in spiking neural networks

**Agnes Korcsak-Gorzo** [1,2,3☉]\*, **Michael G. Müller** [4☉]\*, **Andreas Baumbach** [1,5], **Luziwei Leng** [1], **Oliver J. Breitwieser** [1], **Sacha J. van Albada** [2,6], **Walter Senn** [5], **Karlheinz Meier** [1†], **Robert Legenstein** [4], **Mihai A. Petrovici** [1,5]\*

**1** Kirchhoff-Institute for Physics, Heidelberg University, Heidelberg, Germany, **2** Institute of Neuroscience and Medicine (INM-6) and Institute for Advanced Simulation (IAS-6) and JARA-Institute Brain Structure-Function Relationships (INM-10), Jülich Research Centre, Jülich, Germany, **3** RWTH Aachen University, Aachen, Germany, **4** Institute of Theoretical Computer Science, Graz University of Technology, Graz, Austria, **5** Department of Physiology, University of Bern, Bern, Switzerland, **6** Institute of Zoology, University of Cologne, Cologne, Germany

☉ These authors contributed equally to this work.
† Deceased.
\* a.korcsak-gorzo@fz-juelich.de (AK-G); mueller@igi.tugraz.at (MGM); mihai.petrovici@unibe.ch (MAP)

**Data Availability Statement:** The source code and data to the simulation results and analyses presented in this manuscript are available from https://doi.org/10.5281/zenodo.5512526.

## Abstract

Being permanently confronted with an uncertain world, brains have faced evolutionary pressure to represent this uncertainty in order to respond appropriately. Often, this requires visiting multiple interpretations of the available information or multiple solutions to an encountered problem. This gives rise to the so-called mixing problem: since all of these "valid" states represent powerful attractors, but between themselves can be very dissimilar, switching between such states can be difficult. We propose that cortical oscillations can be effectively used to overcome this challenge. By acting as an effective temperature, background spiking activity modulates exploration. Rhythmic changes induced by cortical oscillations can then be interpreted as a form of simulated tempering. We provide a rigorous mathematical discussion of this link and study some of its phenomenological implications in computer simulations. This identifies a new computational role of cortical oscillations and connects them to various phenomena in the brain, such as sampling-based probabilistic inference, memory replay, multisensory cue combination, and place cell flickering.

## Author summary

Activity oscillations are a ubiquitous and well-studied phenomenon throughout the cortex. At the same time, mounting evidence suggests that brain networks perform sampling-based probabilistic inference through their dynamics. In this work, we present a theoretical and a computational analysis that establish a rigorous link between these two phenomena: background oscillations enhance sampling-based computations by helping networks of spiking neurons to quickly reach different high-probability network states, i.e., computational results.

Such an acceleration of sampling is required for efficient learning and inference in neural networks. Our results show that oscillations provide this acceleration robustly over

**Funding:** The authors gratefully acknowledge funding from the state of Baden-Württemberg through bwHPC (https://www.bwhpc.de) [AKG, AB]; from the German Research Foundation DFG (https://dfg.de) through grant number INST 39/963-1 FUGG (bwForCluster NEMO) [AKG, AB]; from the European Union's Horizon 2020 Framework Programme for Research and Innovation (https://ec.europa.eu/programmes/horizon2020) under grant agreement number 945539 (Human Brain Project SGA3) [AKG, MM, AB, LL, OJB, SvA, WS, KM, RL, MAP], number 785907 (Human Brain Project SGA2) [AKG, OJB, SvA, WS, KM, MAP] and number 720270 (Human Brain Project SGA1) [AKG, OJB, KM, MAP]; from the European Union Seventh Framework Programme ([FP7/2007-2013]) under grant agreement number 604102 (Human Brain Project) [AKG, LL, OJB, KM, MAP]; from the Helmholtz Association (https://helmholtz.de) Initiative and Networking Fund under project number SO-092 (Advanced Computing Architectures) [AKG]; from the Austrian Science Fund FWF (https://fwf.ac.at): I 3251-N33 [MM, RL]; from the Manfred Stärk Foundation [AB, MAP]; and from the Heidelberg Graduate School for Physics (https://hgsfp.uni-heidelberg.de) [LL]. The funders had no role in study design, data collection and analysis, decision to publish, or preparation of the manuscript.

**Competing interests:** The authors of this manuscript have read the journal's policy and declare the following competing interest: Author Karlheinz Meier was unable to confirm his authorship contributions. On his behalf, the corresponding authors have reported his contributions to the best of their knowledge.

different frequency bands and in different network conditions. This suggests a similar functional role of oscillations throughout the cortex. As unspecific background input is enough to evoke this acceleration, our proposed mechanism has a very general scope. We show how such a view on oscillations ties in with a multitude of experimental observations and discuss various opportunities for constraining our model with new experimental data.

Overall, the mechanism we put forward is general and robust and leads to a new understanding of oscillations in the context of sampling-based computations. Our model offers a computational explanation for many related experimental observations that are linked to cortical oscillations.

## Introduction

The ability to build an internal, predictive model of reality endows an agent with a clear evolutionary benefit. How the mammalian brain accomplishes this feat remains a subject of debate, but the representation of uncertainty certainly plays a role, considering the probabilistic nature of sensory data and uncertainty about past and future events. A good representation of an uncertain reality must allow efficient access to a large variety of plausible beliefs about the environmental state.

Distributions over sensory data, characteristic for natural scenes, are complex in the sense that the coexisting beliefs about the data manifest as numerous deep, dissimilar modes of the state space—one of the many facets of the curse of dimensionality. In probabilistic models of such complex data, exact inference becomes intractable, but the distribution can be approximated by sampling. Rapid convergence towards the target distribution requires the sampler to switch (or mix) between these modes frequently. However, due to their dissimilarity, this switching is notoriously difficult for most sampling methods, an issue which is known as the "mixing problem".

In this manuscript, we put forward a hypothesis for how the brain can efficiently overcome this challenge. In doing so, we unify two aspects of cortical dynamics under a common normative framework: spike-based probabilistic inference and cortical oscillations. Both of these phenomena have been well-studied but have not been explicitly linked in the context of spiking neural networks. In particular, we consider the interpretation of spiking activity in the cortex as probabilistic inference via sampling, which has gained ample experimental [1–3] and theoretical [4–8] support over the last decade. Mathematically, these models are closely related to Gibbs sampling, which tends to get stuck in single states of high probability that act as local attractors.

We propose that this problem of sampling-based representations can be overcome by firing rate oscillations. Firing rate oscillations over multiple frequency bands are a naturally emerging phenomenon in spiking networks [9–12] and have been extensively studied in the mammalian brain [13, 14]. Notably, they appear to play an important role both during awake perception [15–17] and during sleep [18, 19], suggesting a fundamental role in cognition and learning. In previous modeling studies, oscillating changes of neuronal excitabilities have been shown to be beneficial for mixing [20–24], but how such changes might arise on the cellular level within networks of spiking neurons has thus far remained unclear.

We propose that the background firing rate of cortical neurons can be interpreted as a (computational) temperature and can accordingly modify the probability landscape sampled by cortical circuits. If the background activity is oscillatory, the network temperature changes

periodically and phase-dependent stationary distributions emerge. By cyclically alternating between "hot" and "cold" periods, cortical networks can effectively instantiate a tempering schedule, with hot phases corresponding to flat probability distributions in which the network can move freely and cold phases representing the multimodal target distribution. This schedule allows networks to escape from local minima and efficiently sample from challenging distributions characterized by multiple high-probability modes separated by large low-probability volumes of the state space.

In this work, we provide an analytical treatment of tempering in spiking networks induced by cortical background oscillations and demonstrate the benefits of this phenomenon in simulations. We explicitly consider current-based and conductance-based synaptic interactions as well as different network architectures and discuss links to experimental data. These observations establish a novel connection between multiple observed cortical phenomena, as well as between these experimental findings and normative theoretical models of brain computation.

## Experiments and results

To understand how cortical oscillations affect computation at the network scale, we study the behavior of single spiking neurons and networks of spiking neurons under variable levels of background activity. We first consider current-based leaky integrate-and-fire (LIF) neurons, for which we can derive analytical expressions for the neuronal response. We show how the level of background input affects the input-output relationship of individual neurons (Section *Single-neuron statistics*). We then discuss the effect of the background activity on entire networks (Section *Temperature in spiking networks*), where we show that this local increase of stochasticity at the single-neuron level gives rise to corresponding changes of the probability landscape at the network level. In particular, we find that these changes can be parametrized by a Boltzmann temperature parameter. Moving to recurrent networks as models of computation in the sensory cortex, we establish a rigorous interpretation of cortical oscillations as a tempering algorithm (Section *Temperature in spiking networks*). We then demonstrate the functional advantages of such oscillation-induced tempering for generative models of the visual hierarchy trained on two different visual datasets (Section *Mixing in high-dimensional multimodal data spaces*). Subsequently, we generalize our findings by lifting several previous assumptions regarding neuron and synapse dynamics and parameters required for mathematical precision. In particular, we extend our simulations to more biologically plausible conductance-based synapses across a range of different parameter regimes (Section *Impact of conductance-based synaptic input*). Using these more general models, we show that, in a sensory disambiguation task, background oscillations have the same effect as in the previous simulations (Section *Background oscillations and behaviorally relevant sampling tasks*), and discuss how sampling models based on oscillatory background input can be linked to experimental data on hippocampal activity within theta cycles (Section *Constraining sampling models with experimental data*).

### Single-neuron statistics

Cortical neurons are embedded in a noisy environment (Fig 1a). In addition to functional input $I_{in}$, their many presynaptic partners provide them with an effectively stochastic background [8, 25]. This background activity leads to stochastic single-neuron behavior [26]. To understand this behavior, we consider a simple LIF neuron model with current-based input synapses (see Section *Neuron models* in Methods). The neuron receives a large number of background inputs, with firing rates $v_i$ and synaptic efficacies $w_i$. In line with standard literature, we model this stochastic background input as uncorrelated Poisson spike trains [27]. We

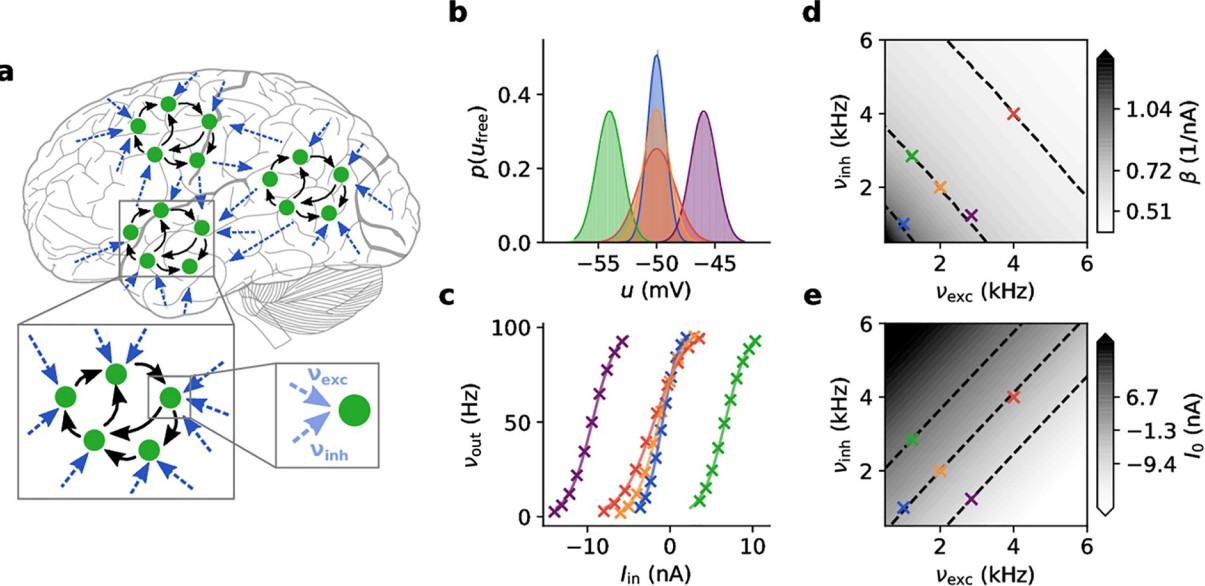

**Fig 1. Response functions of neurons in an ensemble. (a)** Cortical ensemble of networks. The spike input received by a neuron can be partitioned into functional (solid black arrows) and background (dashed dark blue arrows) input. The background can be partitioned into an excitatory and an inhibitory subset (dashed light blue arrows). In the following panels, we consider one such neuron under five different illustrative background regimes, each of which is assigned a specific color. **(b)** Steady-state free membrane potential distributions. Shaded areas: numerical simulation; solid lines: analytical approximation using Eqs 1 and 2. Purple, orange, green: same $\sigma_u$, different $\mu_u$; blue, orange, red: same $\mu_u$, different $\sigma_u$. **(c)** Corresponding neuronal response functions. Crosses: numerical simulation; solid lines: logistic fit with Eq 3. **(d)** Slope parameter $\beta$ and **(e)** offset $I_0$ of response functions under various background regimes defined by their respective pairs of excitatory and inhibitory input rates ($\nu_{\text{exc}}$, $\nu_{\text{inh}}$). Dashed isolines indicate configurations of constant slope (cf. Eq 4) or offset, with specific values given as colorbar ticks. Note the approximate linearity of the contour lines.

first consider the free membrane potential $u_{\text{free}}$ of this neuron, that is, the membrane potential in the hypothetical case that there is no firing threshold. The steady-state distribution $p(u_{\text{free}})$ is well-described by a Gaussian (Fig 1b) with moments

$$\mu_u := \mathbb{E}[u_{\text{free}}] = E_{\text{l}} + \frac{I_{\text{in}} + \sum_i w_i \nu_i \tau_{\text{s},i}}{g_{\text{l}}} \ , \tag{1}$$

$$\sigma_u^2 := \text{Var}[u_{\text{free}}] = \sum_i \nu_i w_i^2 \frac{\tau_{\text{s},i}^2}{2 g_{\text{l}}^2 (\tau_{\text{m}} + \tau_{\text{s},i})} \ , \tag{2}$$

where $E_{\text{l}}$ and $g_{\text{l}}$ are the leak potential and conductance, and $\tau_{\text{m}}$ and $\tau_{\text{s}}$ are the membrane and synaptic time constants (see Section 4.3 in [28]). Here, $\Sigma_i$ runs over all background presynaptic partners. Note that excitatory and inhibitory inputs (defined by the sign of the synaptic weight $w_i$) can cancel each other out in the mean but always add up towards the variance of the free membrane potential distribution.

Upon introducing a firing threshold, some portion of the free membrane potential probability density will lie above it, causing the neuron to spike stochastically. The shape of the neuronal response function, i.e., the firing rate in response to a constant input current $I_{\text{in}}$, depends strongly on the characteristic time constant of the neuronal membrane. Cortical neurons under strong presynaptic bombardment have been shown to operate in a high-conductance regime [29], which greatly reduces the effective membrane time constant $\tau_{\text{m}}$. Under such conditions, the neuronal response function (Fig 1c) can be well approximated by a logistic

function [7]:

$$v_{\text{out}}(I_{\text{in}}) = \frac{1}{1 + \exp(-\beta(I_{\text{in}} - I_0))} \ . \tag{3}$$

Hence, the neuron's stochastic response is characterized by two parameters, the offset $I_0$ and the slope $\beta$ of the sigmoid. Both of these depend on the background activity. The response function can be intuitively understood as the area under the free membrane potential distribution that lies above the firing threshold. Thus, its shape is similar to the integral of $p(u_{\text{free}})$, its offset $I_0$ has a similar linear dependence on $\mu_u$, and its slope parameter $\beta$ will decrease for increasing $\sigma_u$. Their exact dependence on the background rates is shown in Fig 1d and 1e. In particular, the relationship between the slope of the response function and the standard deviation of the free membrane potential distribution is well-approximated by a linear function, which allows us, in turn, to establish the relationship between the slope parameter $\beta$ and the total (i.e., summing over all background presynaptic partners) excitatory and inhibitory background firing rates $v_{\text{exc}}$ and $v_{\text{inh}}$ and the corresponding weights $w_{\text{exc}}$ and $w_{\text{inh}}$ using Eq 2:

$$\frac{1}{\beta} \propto \sigma_u \propto \sqrt{w_{\text{exc}}^2 v_{\text{exc}} + w_{\text{inh}}^2 v_{\text{inh}}} \ . \tag{4}$$

To summarize, we have established how the stochastic response of individual LIF neurons depends on the level of background input. In particular, the background input determines the slope $\beta$ of the (logistic) neuronal response function.

## Temperature in spiking networks

As discussed in Section *Single-neuron statistics*, under Poisson background activity, individual neurons react to their input stimulus in a well-defined stochastic manner. Based on this result, we show here how the level of background activity influences the stochastic properties of a recurrently connected network of LIF neurons (Fig 2a).

In a spiking network, the information conveyed by a neuron at any point in time can be described as binary: the neuron either spikes or it does not. A spike has a twofold effect: it initiates a refractory period and elicits postsynaptic potentials (PSPs) in postsynaptic partner neurons. We can therefore view the binary state $z$ of a neuron being refractory ($z = 1$) or not ($z = 0$) following a spike as corresponding to the state communicated to its downstream partners ([4, 7]; see Fig 2a). Thus, each neuron can be interpreted as sampling from the conditional distribution $p(z_k = 1 | z_{\backslash k})$, i.e., the probability of the $k^{th}$ neuron to be in the state "1" given the states of all other neurons $z_{\backslash k}$.

In general, the joint distribution sampled by the network cannot be given in a closed form. To allow an analytical approach, we begin with a set of assumptions about the neuron and network parameters (see Section *Neuron models* in Methods and Section *Entropy of spiking sampling networks* in Methods), but later show that they can be relaxed without affecting the computational network properties discussed here. For parameters emulating a high-conductance state [29] the activity of an LIF network can be interpreted as sampling from a joint Boltzmann distribution [7]

$$p_T(\mathbf{z}) \propto \exp(-E(\mathbf{z})/(k_B T)) \ , \tag{5}$$

where $E(\mathbf{z}) = -\frac{1}{2}\sum_{kj} W_{kj} z_k z_j - \sum_k B_k z_k$ represents the energy of a particular joint state $\mathbf{z}$, with $W_{kj}$ denoting effective recurrent synaptic weights and $B_k$ effective individual neuron biases. Here, $k_B$ is the Boltzmann constant and $T$ is the ensemble (Boltzmann) temperature. For such

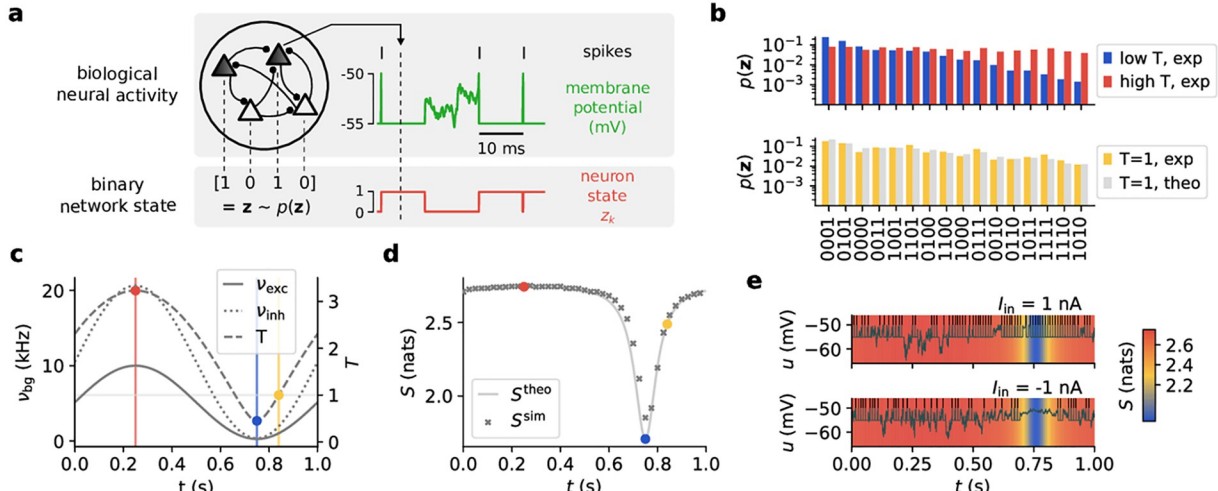

**Fig 2. Effects of oscillatory background activity on sampled distributions.** (a) Network dynamics. Each neuron encodes a binary random variable according to its refractoriness. When the membrane potential (green) is clamped to the reset value, the neuron state (red) is considered to be $z = 1$ ($z = 0$ otherwise). The collection of the resulting network states $\mathbf{z}$ forms an estimate for the implemented probability distribution $p(\mathbf{z})$. (b) Distributions sampled by a 4-neuron network at the three temperatures marked in (c). States are ordered according to their respective probabilities at the low temperature to emphasize the effect of tempering visually. (c) Time course of excitatory and inhibitory background rates (dashed and dotted lines, Eq 10), along with the associated temperature (solid line, Eq 4). Note that $\nu_{\text{exc}}$ is scaled by 0.5 and $w_{\text{exc}}$ by the square root of the inverse scaling factor to demonstrate that balance is independent of such a rescaling. (d) Simulated (crosses) vs. calculated (solid line) entropy course $S(t)$. The slight lag is due to the finite relaxation time constants $\tau_s$, $\tau_{\text{ref}}$ of the network (Eq 35 only holds strictly for quasi-static temperature changes). (e) Effect of tempering on individual membrane potentials and spiking activity. The background color represents the corresponding entropy.

a network state distribution, the state probability of each neuron $k$ is given by [4, 28]

$$p(z_k = 1|\mathbf{z}_{\backslash k}) = \frac{1}{1 + \exp\left(-\frac{\sum_{j \neq k} W_{kj} z_j + B_k}{k_{\text{B}} T}\right)} . \tag{6}$$

Note that this equation has the same form as the neuronal response function in Eq 3.

Since weights and biases can both be interpreted as movements along the $I_{\text{in}}$-axis of the neuronal response function, their simultaneous multiplicative scaling by $T$ is equivalent to a horizontal stretching of the response function. This similarity allows us to identify

$$\beta = 1/(k_{\text{B}} T) , \tag{7}$$

again analogous to statistical physics, for a Boltzmann constant $k_{\text{B}}$ that relates the (unitless) reference temperature $T = 1$ to a chosen set of neuron and background parameters via the resulting response function (here, the unit of $k_{\text{B}}$ is nA). Note that the Boltzmann parametrization with unitless weights $W_{kj}$ and biases $B_j$ in Eq 6 is different from the synaptic weights and biasing effects in Eq 3 induced by leak, threshold potentials, unbalanced input etc. in the LIF domain, but they can be linearly mapped such that the sampling distributions match (see Eqs 33 and 34 in Methods for details). Eqs 4 and 7 thus establish an exact relationship between the ensemble temperature and the background firing rates:

$$T \propto \sqrt{w_{\text{exc}}^2 \nu_{\text{exc}} + w_{\text{inh}}^2 \nu_{\text{inh}}} . \tag{8}$$

In order to study the effect of pure temperature variations without affecting neuronal offsets, excitatory and inhibitory background rates need to be balanced. Such a balance is also well-documented in vivo [30, 31]. Note that this is not simply achieved by setting $w_{\text{exc}}\nu_{\text{exc}}\tau_{\text{exc}}$ $= w_{\text{inh}}\nu_{\text{inh}}\tau_{\text{inh}}$ and thus effectively equalizing the effects of excitation and inhibition; while this would leave $\mu_u$ unchanged, it would still affect $I_0$ (cf. Fig 1b and 1c). A balanced regime can be achieved by a linear dependence between firing rates (Section *Spike response of sampling neurons* in Methods), following one of the isolines in Fig 1e, which are well approximated by

$$\nu_{\text{inh}} = \nu_0 + m\nu_{\text{exc}} . \tag{9}$$

The exact parameters $\nu_0$ and $m$ that are necessary for balance depend on the synaptic time constants and background input weights (see Section *Temperature as a function of background rates* in Methods). Following such an isoline then results in a constant $I_0$ and a $\sqrt{\nu}$ dependence of the (inverse) slope parameter $1/\beta$ (see Section *Spike response of sampling neuron* in Methods). While this approach enables a strict realization of a Boltzmann temperature, the achieved effect does not rely strongly on such a balance, as we discuss below (Section *Impact of conductance-based synaptic input*). Following Eq 4 we can maintain the balance if we rescale the $\nu_{\text{exc}}$ and multiply $w_{\text{exc}}$ by the square root of the scaling factor, which we apply in Fig 2.

With this definition of temperature, we now turn to its effects on the distribution. In Eq 6, the ensemble (Boltzmann) temperature $T$ scales the effective weights and biases multiplicatively, identically to its effect in statistical physics: as the temperature of an ensemble rises, particle interactions (here: synaptic weights) and external fields (here: neuronal biases) become increasingly inconsequential.

We can observe a similar effect on the sampled distribution when modulating the temperature implemented by the background input (see Fig 2b): at high temperatures, the distribution becomes flat, while at low temperatures, the high-probability maxima become even more pronounced. Cyclic heating and cooling—enabled here by oscillatory background—can thus alternate between hot phases with equalized state probabilities and cold phases for reading out the most relevant samples of the correct distribution, where the sampled distribution approximates the target distribution most closely in the $T = 1$ crossings (see S1 Fig for the divergence during one cycle). Such a cycle is often referred to as tempering. We consider a simple sinusoidal oscillation as a basis function for modeling cortical oscillations:

$$\nu_{\text{exc}}(t) = \frac{\nu_{\text{max}} - \nu_{\text{min}}}{2}\sin(2\pi f_{\text{osc}}t) + \frac{\nu_{\text{max}} + \nu_{\text{min}}}{2} , \tag{10}$$

with minimum rate $\nu_{\text{min}}$, maximum rate $\nu_{\text{max}}$, and oscillation frequency $f_{\text{osc}}$. This time course implicitly also defines $\nu_{\text{inh}}(t)$ through Eq 9, such that in this setup, excitation and inhibition vary synchronously (see Fig 2c), as observed in vivo (see, e.g., [32]). Note that the network activity follows the instantaneous level of balanced background input (S2 Fig).

The resulting temperature thus also varies periodically, with the square root of a sine (see Fig 2c and Eq 8). Moreover, the ensemble temperature controls the entropy of the sampled distribution, which effectively describes the "disorder" of the network and corresponds to the uniformity of the sampled distribution. For higher temperatures, as the sampled distribution becomes more uniform, the entropy increases (Fig 2d). In high-temperature/high-entropy states, membrane potentials are extremely noisy, causing neurons to fire randomly and independently. In contrast, in low-temperature/low-entropy states, membrane potentials are nearly constant, and neurons are "frozen" in certain states, firing either persistently or not at all (Fig 2e).

## Mixing in high-dimensional multimodal data spaces

In the following, we discuss the computational role of background oscillations for spiking networks trained to represent complex distributions over high-dimensional visual data. Here, we have chosen two commonly used visual datasets to serve as examples, but our conclusions hold for arbitrary distributions. As a simplified model of cortical visual hierarchy, we consider recurrent layered spiking networks consisting of LIF neurons, which we train as simultaneous generative and discriminative models (Fig 3a). These two forms of computation happen concurrently and bi-directionally: the label neurons classify the state of the visible layer, while the visible neurons adapt their states to produce images that are compatible with the class represented by the label layer. For each class, during the preceding training, probability mass was built up in the corresponding region of the probability landscape, forming the modes of the network.

High-dimensional but well-recognizable visual data confronts such networks with two contradictory challenges. On the one hand, they need to produce good samples, i.e., clean images corresponding to particular sharp high-probability modes separated by large vanishing-probability volumes of the state space that correspond to out-of-distribution samples. On the other hand, they need to be able to switch between different modes in order to sample from the target distribution fully; this is at fundamental odds with the probability landscape described

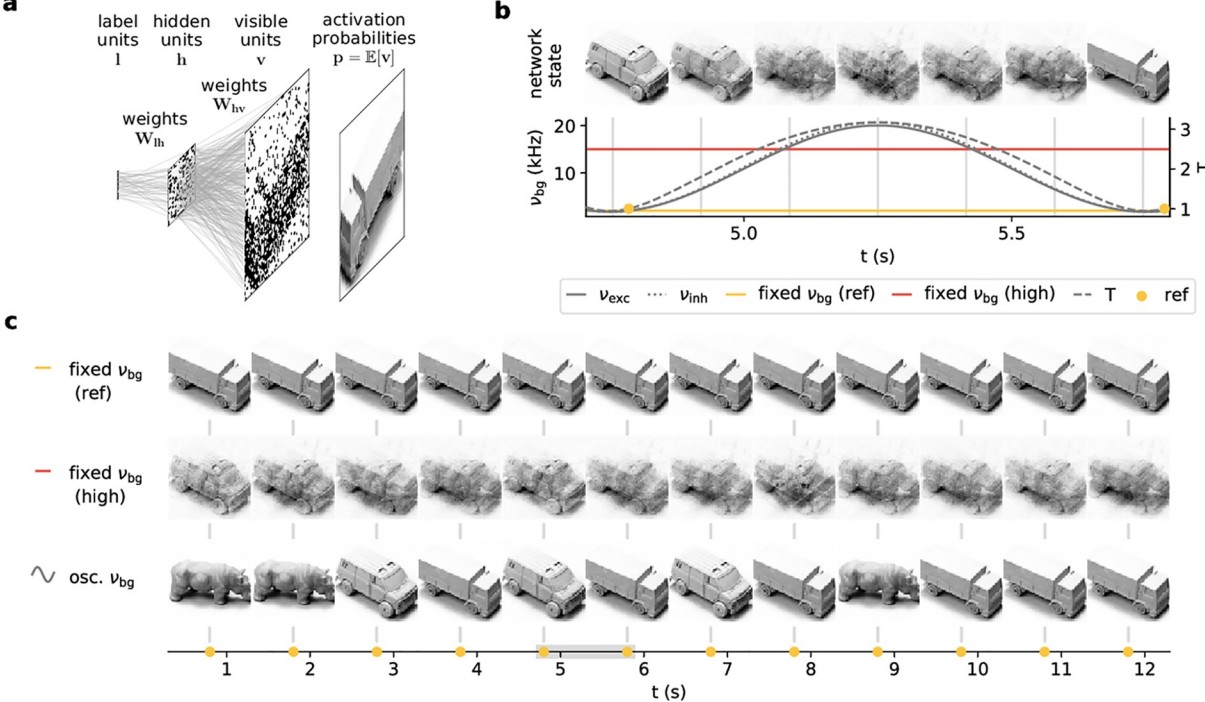

**Fig 3. Background oscillations improve generative properties of spiking sampling networks. (a)** Architecture of a hierarchical 3-layer (visible **v**, hidden **h** and label **l**) network of LIF neurons and example layerwise activity. For a better representation of the visible layer statistics, we consider neuronal activation probabilities $p(v|h)$ rather than samples thereof, to speed up the calculation of averages over (conditional) visible layer states. Here, we show a network trained on images from the NORB dataset. **(b)** Evolution of the activation probabilities of the visible layer (top) over one period of the background oscillation (bottom). **(c)** Evolution of the visible layer over multiple periods of the oscillation compared to a network with constant background input at the reference rate (2 kHz, top) and at a high rate (10 kHz, middle), cf. also yellow and red lines in (b). The activation probabilities are shown whenever the reference rate (see panel b) is reached. The gray bar denotes the period shown in (b).

above. This so-called mixing problem is well-known and quasi-ubiquitous for sampling models.

One solution to this problem was proposed by [33] in the context of Markov-chain Monte Carlo sampling for Ising models, which is intimately related to our form of spike-based sampling in both dynamics and sampled distribution [7]. This simulated tempering method describes a cyclic heating and cooling schedule reminiscent of the periodic temperature variation induced by cortical oscillations discussed above (Eq 10). In-between readouts at the reference temperature, a temporary rise in temperature flattens the probability landscape, allowing the network to escape from local attractors. Thus, Eqs 4, 5 and 10 establish a rigorous analogy between simulated tempering and cortical oscillations, which thereby take on the computational role of enabling mixing in challenging real-world scenarios.

To evaluate these effects, we considered two example scenarios based on well-studied visual datasets: NORB [34] and MNIST [35]. Network training was done using a variant of wake-sleep learning [36], a contrastive Hebbian scheme inspired by biological phenomenology and widely used for sampling models (see in particular [37]). A background rate of $\nu_{\text{exc}} = \nu_{\text{inh}} = 2$ kHz was chosen as reference, implicitly defining the reference temperature $T = 1$.

For visual datasets, the weakened correlations at higher temperatures correspond to blurred images. For the network trained on NORB, this is particularly well observable (cf. Fig 3b). The network produces sharp images at low background rates, whereas the images become blurred under increased background activity. Note especially how the network enters a superposition of several "clean" states at higher background rates. Constant background stimulus cannot reproduce the ease of switching between different image classes (modes). The network is either stuck in one mode while producing sharp images ($T = 1$ upper row in Fig 3c) or only able to produce blurred images ($T = 2.5$ middle row in Fig 3c). Tempering through background oscillations effectively combines these two regimes, allowing a better sampling of the target distribution at phases where the reference temperature is reached (lower row in Fig 3c).

The effectiveness of this tempering schedule depends on the parameters of the background oscillations: $\nu_{\text{min}}$, $\nu_{\text{max}}$, and $f_{\text{osc}}$. In particular, the frequency $f_{\text{osc}}$ plays a critical role, as it represents a tradeoff between exploration and exploitation of the network's state space. Low frequencies guarantee that the network has time to relax towards its momentary stationary distribution $p_T$, with $f_{\text{osc}} \to 0$ representing the quasi-static limit, i.e., constant background. This enables accurate sampling from the target distribution at $T = 1$, as the network loses memory of previous states occupied at higher temperatures. However, lower oscillation frequencies come at the cost of slower sampling, as they increase the time between consecutive readouts. Furthermore, frequencies significantly lower than 0.1 Hz are rarely observed in vivo [14]. In the following, we study the behavior of spiking sampling networks under different background oscillation regimes for a network trained on handwritten digits from the MNIST dataset.

Two essential quality criteria for any sampling network are its mixing speed and sample fidelity. In principle, Eq 5 allows an analytical evaluation of these properties, but in practice, this is unfeasible for high-dimensional distributions. We, therefore, use a sample-based measure, the indirect sampling likelihood (ISL, see [38]). The ISL accumulates fidelity values for all generated samples, assigning high values if they are similar to images in the test set and low values otherwise. Additionally, the rate at which the ISL increases over time implicitly represents a measure of the mixing speed. We use the distribution of times between label switches as a more explicit measure of mixing times for different image categories.

Our MNIST-trained network allows a quantitative evaluation of the benefits of oscillation-induced tempering. In each tempering cycle, around $T = 1$, one digit stabilizes in the visible

layer for a wide time window (see S3 Fig). The corresponding network mode is defined by the label neuron with the highest probability inferred from the hidden layer activity. With oscillatory background (Fig 4a), the sampled digits and labels change more frequently as compared to constant background (Fig 4b). Consequently, the average mode duration, as defined by the time interval between two mode switches, is shorter for oscillatory background (compare Fig 4c and Fig 4d). Since frequent mode switches are essential to efficiently cover the target distribution, the Kullback-Leibler divergence (DKL) between the target and sampled distribution also decreases more rapidly with oscillatory background (Fig 4e). Furthermore, the ISL converges to higher values compared to the constant background (Fig 4f), which indicates an overall better tradeoff between generating clear examples of the imprinted classes and good mixing between these classes (also see S4 Fig). Note that tempering can likewise improve the network's performance in inference tasks (S5 Fig). For details to DKL, ISL, and the network setup, see Section *Image generation examples: NORB and MNIST* in Methods.

Next, we studied tempering under a range of biologically plausible regimes, with background rates (per neuron) varying between 0.5 and 30 kHz and oscillation frequencies ranging from the alpha range to the first slow-wave band [13]. In the landscapes over the mode durations (Fig 4g) and the ISLs (Fig 4h), we find that the most important prerequisite for effective tempering is the maximum background rate, as the temperature between readouts has to be

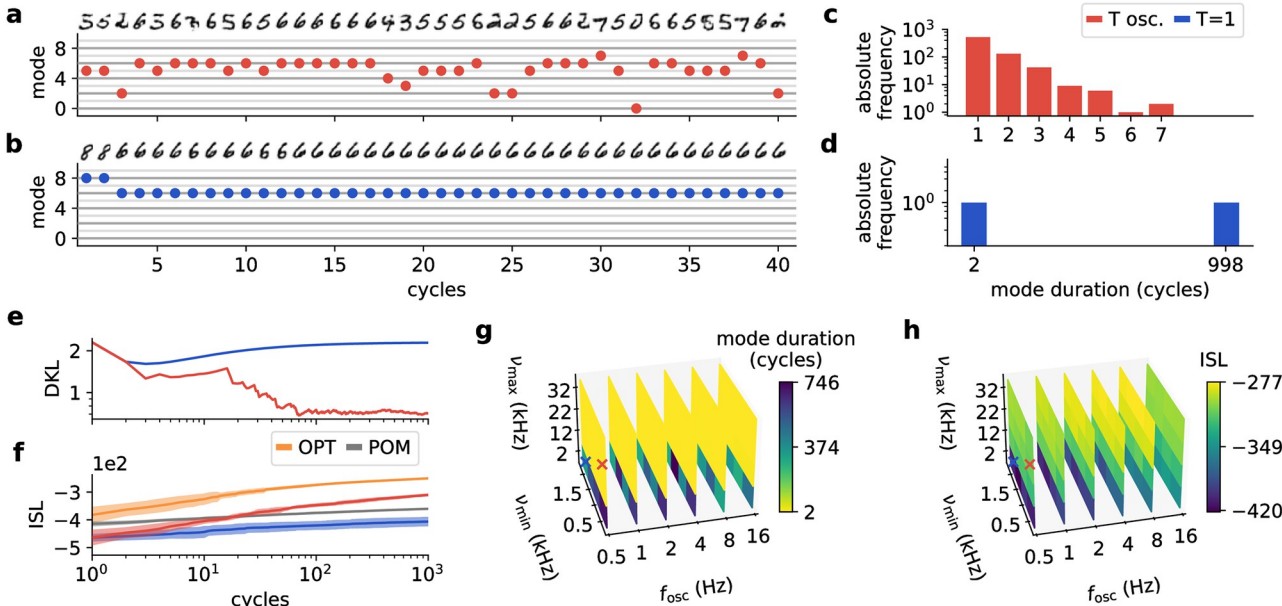

**Fig 4. Parameter dependence of tempering effectiveness. (a, c)** Visible and label layer activity of an LIF network trained on the MNIST dataset, with **(b,d)** showing the corresponding mode duration distributions (the active mode corresponds to the image class and is determined by the most active label neuron). The network with oscillatory background (red) moves quickly between modes, with correspondingly short mode durations, whereas the network with constant background activity (blue) switches to the "6" mode after two samples and remains there until the last of the $10^3$ collected samples. **(e)** Kullback-Leibler divergence (DKL) between the distribution of sampled modes and the uniform distribution. The sampled distribution quickly becomes significantly more uniform for the oscillatory (red) compared to the constant (blue) background. **(f)** Indirect sampling likelihood (ISL) as a measure of image quality and diversity for the two background settings and, for orientation, for the optimal sampling (OPT, orange) and the product of marginals (POM, gray). Under this measure, the averaged MNIST images described by the POM are more similar to the entire dataset than the near-unimodal distribution generated under constant background at $T = 1$. Similarly, the network with oscillatory background needs several samples to produce a distribution that is diverse enough to overtake the POM. The mean (solid lines) and standard deviation (shades) over 10 runs of $10^3$ samples are plotted. **(g)** Average mode duration for different oscillation parameters: The peak background rate $v_{max}$ represents the most critical parameter and needs to be high enough to enable good mixing. The minimum background rate $v_{min}$ and the oscillation frequency $f_{osc}$ are less important. **(h)** Same as (g) for the ISL values. The image quality remains consistently high across a wide range of parameter configurations. The data used for (a-f) corresponds to the simulations marked by the red and blue crosses, respectively. Values represent averages over 10 runs of $10^4$ samples.

high enough for frequent mode switches. For our networks, this required input rates above 10 kHz (Fig 4g and 4h). On the other hand, the minimum background rates in the cold phases have a much smaller influence. In general, effective tempering is achieved over a wide range of oscillation parameters (yellow and light green areas in Fig 4g and 4h) covering all studied frequency bands. Overall, the best performance was achieved in the slow-wave regime.

## Impact of conductance-based synaptic input

Up to this point, we have used a mathematically tractable model providing an exact link between background input and network behavior. This link leads to a clear interpretation of background oscillations as a schedule of temperature changes within networks of neurons. We now show that these conceptual results hold over a wider range of neuron and synapse models with different parameters.

To this end, we relax the assumptions of the previously considered current-based models in three ways. First, we use conductance-based synaptic interactions, which are known to be a good description of the behavior of biological neurons but prevent an exact analytical treatment. Second, we drop the assumption that the response functions are tuned such that sampling is unbiased (see above) and consider the general case where the response behavior changes as the background input rates are varied. Third, we consider a range of physiological parameter settings, including different ways in which excitatory and inhibitory rates vary over the oscillatory cycle. By exploring different parameter regimes with few assumptions, we highlight that the effect of oscillating background input described above holds regardless of the specific model details and can be expected to shape sampling computations in various brain networks. As before, we first investigate the properties of individual neurons before moving on to network-level effects.

**Background input scenarios.**   The behavior of neurons with conductance-based synapses under synaptic bombardment, in general, differs from the current-based case. In particular, let us first consider the variance of the membrane potential as a function of the background input rate and the synaptic efficacies of the background input synapses, see Fig 5a and 5b. As the growth of this variance underlies the increase of the temperature (see above), this is an important indication of the sampling behavior of the neuron. Fig 5a shows that the variance of the membrane potential grows monotonically with the rate of the background input for current-based synapses. Interestingly, this is not the case with conductance-based background synapses (Fig 5b, see also Eq 21 in Section *Conductance-based LIF model* in Methods). This raises the question of whether the same, simple relationship between background input rate and sampling temperature is present in the conductance-based case. However, neuron response functions still depend monotonically on the input rates, as the relative impact of inputs is also weakened by an increasing total conductance. In the following, we show that, as a result, the influence of background input rates on the effective temperature using conductance-based neurons matches the influence in the current-based case.

As for current-based neurons (Eq 9), we investigate the impact of the background input rate by co-varying the excitatory and inhibitory input rates $v_{\text{exc}}$ and $v_{\text{inh}}$ in a linear manner using a scaling parameter $\alpha$ such that

$$v_{\text{exc}} = \alpha \, v_{\text{exc},1} + v_{\text{exc},0} \qquad \text{and} \qquad v_{\text{inh}} = \alpha \, v_{\text{inh},1} + v_{\text{inh},0} \qquad (11)$$

where $v_{\text{exc},1}$ and $v_{\text{inh},1}$ are some excitatory and inhibitory base rates, and $v_{\text{exc},0}$ and $v_{\text{inh},0}$ are rate offsets.

By different choices of the base rates $v_{\text{exc},1}$, $v_{\text{inh},1}$, rate offsets $v_{\text{exc},0}$, $v_{\text{inh},0}$, together with choices for the excitatory and inhibitory synaptic efficacies of the background input, we obtain

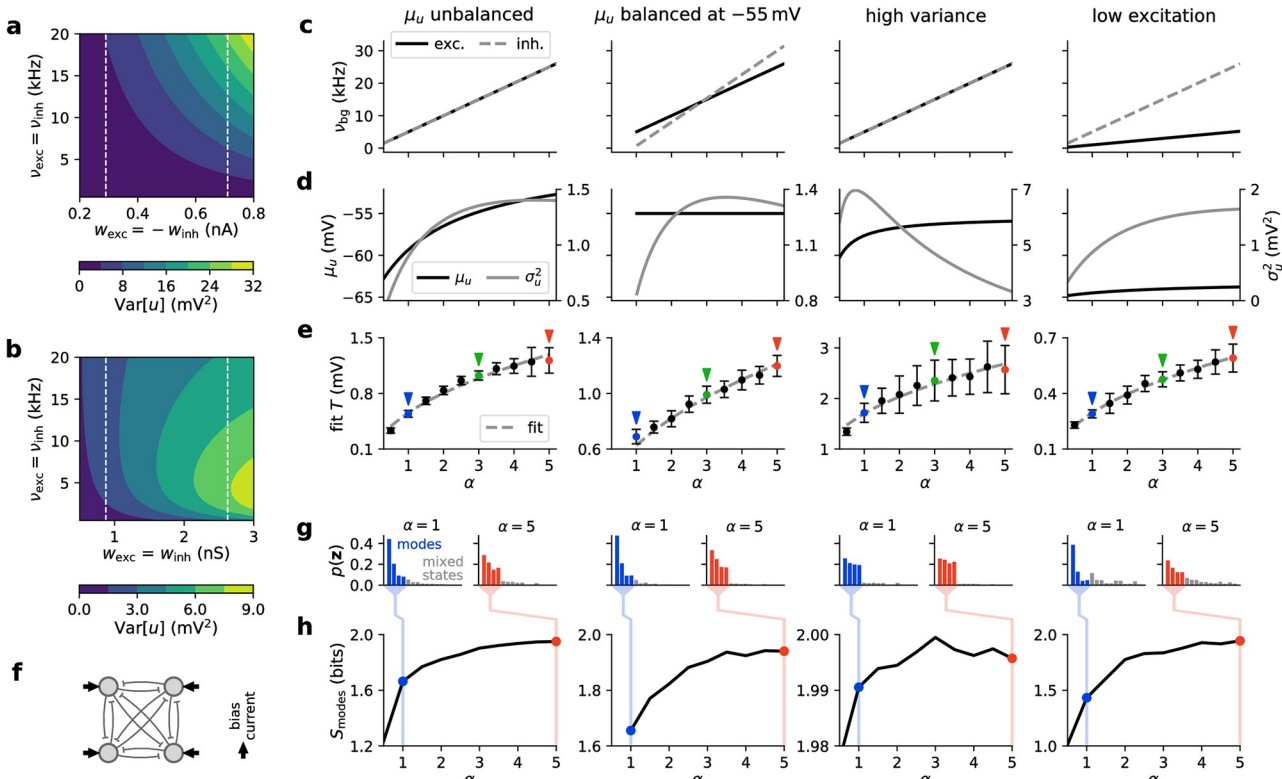

**Fig 5. Background input rate sets the sampling temperature for conductance-based LIF neurons. (a, b)** Variance of the free membrane potential for (a) current-based and (b) conductance-based synaptic interactions as a function of background rate and strength under balanced input rates and weights. Note that $\mathrm{Var}[u]$ is always monotonic in $\nu_{\mathrm{exc}}$ for current-based synapses, which does not hold for the conductance-based case (see the change of $\mathrm{Var}[u]$ along the dashed white lines). **(c–e)** The following panels show four different biologically interesting scenarios of conductance-based background input (ordered in columns). In each scenario, excitatory and inhibitory input is varied in different ways depending on a scaling parameter $\alpha$ (see Eq 11). Furthermore, the synaptic weights for background input are different in each scenario (see Section *Conductance-based input scenarios* in Methods). From left to right: $\mu_u$ unbalanced (one of many possible settings in which $\mu_u$ increases), $\mu_u$ balanced at −55 mV (with constant mean membrane potential close to the firing threshold), high variance (same as the first scenario but with larger synaptic conductances, resulting in a higher variance of the membrane potential), and low excitation (with a lower increase in excitation compared to inhibition). **(c)** Excitatory and inhibitory background input rates as a function of the scaling value $\alpha$. Note that for the $\mu_u$ balanced at −55 mV scenario the balance can only be achieved for $\alpha \geq 1$. **(d)** Mean and variance of free membrane potential resulting from background input scaled by $\alpha$. **(e)** Temperature values resulting from fitting. Dots denote mean, whiskers standard deviation over $N = 50$ independent runs (see Section *Fitting of stochastic models* in Methods for details). Importantly, the temperature increases monotonically with $\alpha$ in all cases. The dashed gray line shows fits according to the theoretical prediction (Eq 4), closely matching mean values. The highlighted values correspond to low (blue, $\alpha = 1$), medium ($\alpha = 3$) and high (red, $\alpha = 5$) background input rates. See Section *Fitting of stochastic models* in Methods for a visualization of the resulting stochastic models and the soft threshold values $u_T$. **(f)** Schematic drawing of the simple network used to illustrate the effect of changing background input rates on the distribution of network states. The network consists of four neurons connected with lateral inhibition, resulting in a network state distribution with four distinct modes (one neuron is active while the others remain silent). Each neuron receives a bias current input (arrows), the amplitude of which defines the probability of the mode corresponding to this neuron being active. See Section *Illustrative sampling task for conductance-based networks* in Methods for details. **(g)** Network state probabilities for low ($\alpha = 1$) and high ($\alpha = 5$) temperatures in a network simulation. Modes are shown in color; mixed states are shown in gray. At high temperatures, the distribution generally becomes flatter. **(h)** Entropy of modes at each value of $\alpha$ (in bits, calculated using Eq 35). The entropy increases as the mode distribution becomes flatter (i.e., closer to a uniform distribution). Note that while the temperature increases in all scenarios (see (e)), the values and changes of the entropy can vary dramatically.

different scenarios that reflect the diversity and complexity of conductance-based interactions (see below). We consider $\alpha \in [0.5, 5]$, which results in background excitatory and inhibitory input rates $\leq$ 30 kHz given the base and offset rate values for the different scenarios. These values are similar to the rate range used in the current-based simulations above and represent a reasonable assumption for cortical neurons as they typically have a large number of

presynaptic partners [39]. Using the input rates and the synaptic parameters, it is possible to calculate the mean $\mu_u$ and variance $\sigma_u^2$ as a function of the background input scaling factor $\alpha$ ([28], Section 6.5; [40]).

The four different scenarios (Fig 5c–5e) considered here are:

- $\mu_u$ *unbalanced:* A general case of realistic synaptic conductances not tuned to any specific regime with equal excitatory and inhibitory background rates ($v_{\text{exc},1} = v_{\text{inh},1} = 5$ kHz and $v_{\text{exc},0} = v_{\text{inh},0} = 0$ kHz). Although $v_{\text{exc}} = v_{\text{inh}}$ in this case (Fig 5c, first column), excitatory and inhibitory inputs are not balanced as the synaptic time constants differ (see Section *Conductance-based LIF model* in Methods). As a result, both $\mu_u$ and $\sigma_u^2$ increase over the considered range of $\alpha$ (Fig 5d, first column).

- $\mu_u$ *balanced at −55 mV:* This scenario mimics the regime of cortical up-states, where neurons have membrane potentials close to the firing threshold (here: $u_{\text{th}} = -50$ mV). Balancing neurons in this fashion, i.e., keeping $\mu_u$ at −55 mV regardless of the value of $\alpha$, can be achieved with specific choices of $v_{\text{inh},1}$ and $v_{\text{inh},0}$ (i.e., $v_{\text{inh}} \neq v_{\text{exc}}$, see Fig 5c, second column, and Eq 40 in Methods). Note that this balancing requires minimal background input, limiting $\alpha$ to $\alpha >$ 1. Here, the variance $\sigma_u^2$ first increases before reaching a peak and slowly decreasing again (Fig 5d, second column).

- *High variance:* This case is similar to the first case above but with larger synaptic conductances for the background input. Larger synaptic conductances give rise to a different regime, which is approximately balanced (small change of $\mu_u$ for changing $\alpha$), but differs from the previous scenarios in two significant ways: first, the overall variance is much higher, and second, the variance decreases with $\alpha$ (Fig 5d, third column). This scenario uses $v_{\text{exc}} = v_{\text{inh}}$ (as the first scenario).

- *Low excitation:* It is unclear whether in the brain excitatory and inhibitory input levels are similar, in particular within oscillations, and it has previously been suggested that oscillations mostly affect inhibition [20]. We, therefore, also consider a scenario in which the inhibitory rates increase much more strongly than the excitation (Fig 5c, last column). This results in a marginal effect on the mean free membrane potential, while the variance increases with $\alpha$ (Fig 5d, last column).

All parameters for the different scenarios are given in Section *Conductance-based input scenarios* in Methods.

**Fitting stochastic models to quantify temperature changes.** To gain an understanding of the stochasticity induced by background activity, we fitted stochastic neuron models to data produced by conductance-based LIF models with background input. This method allows quantifying behavior changes regardless of the precise input conditions and neuron parameters, thus making it possible to describe the sampling temperature even when an analytical treatment is not possible.

To this end, we used the fitting method proposed by [41] to fit a stochastic neuron model with an exponential escape rate function to the behavior of LIF neuron at the given background input scenario. Models with an exponential escape rate were shown to match the firing behavior of cortical pyramidal cells [41] as well as the behavior of simple neuron models when subjected to background input [42]. Furthermore, an exponential escape function is commonly used in theoretical sampling models [4] and is equivalent to the sigmoidal activation function used in the current-based models. The stochastic model is identical to the LIF neuron model described above except for a stochastic firing criterion with instantaneous firing

intensity

$$\rho(u) = \frac{1}{\Delta t}\exp\left(\frac{u - u_{\mathrm{T}}}{T}\right) , \tag{12}$$

where $T$ is the temperature, $u_{\mathrm{T}}$ is the soft threshold (i.e., the value of $u$ where the firing intensity reaches $1/\Delta t$), and $\Delta t$ is the resolution of the discrete-time simulation.

We performed this fitting procedure for different values of $\alpha$ to examine how the fitted model parameters change as the background input strength is varied. For every fit, we used a number of presynaptic spike trains to excite the LIF neuron (see Section *Fitting of stochastic models* in Methods) and recorded the firing times. The resulting firing rates varied markedly (see Section *Fitting of stochastic models* in Methods), highlighting the different operating regimes induced by different levels of background input. We fitted exponential firing intensities to the data (Eq 12). The temperature values resulting from the fitting procedure are shown in Fig 5e.

We found that in all four scenarios, $T$ increases with $\alpha$, even when the variance of the membrane potential plateaus or decreases. As shown above, $T$ grows with $\sqrt{\alpha}$ in the current-based case. Fitting the mean temperatures of the fitted models shows that this relationship also describes the change of the temperature very well in the conductance-based case (Fig 5e, dashed gray lines), i.e.,

$$T \propto \sqrt{\alpha} . \tag{13}$$

These results confirm the role of background rates as an effective ensemble temperature in conductance-based networks with diverse properties, and this method can be used to quantify the temperature changes. Depending on the parameters of the background activity, the covered temperature range can vary significantly, and for realistic parameters, the maximum temperature is limited. This limitation marks a difference to the current-based case, where, in principle, arbitrarily high temperatures can be reached even with relatively low background input rates. Nevertheless, as shown in the following, these temperature changes have important functional consequences at the network level.

## Entropy in networks with conductance-based synapses

To confirm that these changes of the stochastic behavior of single neurons result in changes of the sampling behavior on the network level, we investigated a simple network of conductance-based LIF neurons. We used a network consisting of four neurons with a winner-take-all structure, i.e., each neuron had lateral inhibitory connections to the other neurons (see Fig 5f and Section *Illustrative sampling task for conductance-based networks* in Methods). Winner-take-all structures are of particular interest because they are a common cortical motif [43]. Due to inhibitory competition between the four neurons in the studied network, its probability landscape exhibits four distinct and separated modes, each of which corresponding to one of the four neurons being active while most of the other neurons remain silent. Each neuron was injected with a constant individual current. The strengths of these currents were different for different neurons, leading to different probabilities for the four modes. Furthermore, each neuron received background input, controlled by setting $\alpha$ according to the different scenarios.

This setup allowed us to test the changes of the probability landscape when the background input strength changes. We found that in every scenario, the mode distribution becomes more uniform for high levels of background input (Fig 5g). To quantify the changes, we calculated the entropy of the mode probabilities (Fig 5h). In each scenario, the entropy increases with $\alpha$,

indicating an increase in the sampling temperature. In the high-variance case, the effect is small but shows the same trend (larger entropy for larger $\alpha$), which is surprising as the variance of $u$ decreases as $\alpha$ is increased. This can be explained as follows: even as the variance decreases, the overall synaptic conductance evoked by background input grows. Therefore, as $\alpha$ increases, the effect of the background input grows stronger relative to the input from the recurrent network connections, thus leading to more equal responses.

This experiment confirms that changes in the background activity of conductance-based networks give rise to the same qualitative phenomena as in the current-based case. We next show the relevance of this effect in a behaviorally relevant sampling task.

## Background oscillations and behaviorally relevant sampling tasks

As in networks of current-based neurons, we expect that background oscillations structure computations into distinct phases when using conductance-based networks, which we investigate next. In contrast to the current-based sampling experiments, we do not restrict ourselves to the precise conditions required for unbiased sampling and instead consider the more general case of arbitrary parameters.

The link between activity levels and sampling temperature described previously suggests that brain networks alternate between sampling at high temperatures, allowing rapid traversing of the state space for good mixing, and sampling at low temperatures, promoting convergence to states of high probability. We investigated this effect using a stimulus disambiguation task, in which a network was required to find coherent interpretations for conflicting inputs across three different sensory modalities (auditory, visual, and somatosensory, Fig 6a). Each sensory modality (vertical columns in Fig 6a) was represented by a group of three neuronal assemblies, with each of these assemblies encoding one of three possible interpretations of the input. To represent mutually exclusive interpretations, the assemblies representing each sensory modality (boxes in Fig 6a) had lateral inhibitory connections, instantiating a winner-take-all (WTA) network. Assemblies representing the same interpretation across different sensory modalities were set up as mutually and recurrently excitatory (solid lines in Fig 6a). Thus, when such a triplet of assemblies across sensory modalities was active, the network encoded a coherent interpretation of the input. We injected a small bias current into neurons in three assemblies: the assembly encoding interpretation #1 for the auditory modality, the assembly for interpretation #2 for the visual modality, and the assembly for interpretation #3 for the somatosensory modality. As these bias currents were identical, none of the three competing interpretations was favored, thus leading to sensory ambiguity. To correctly represent such an ambiguous situation, the network is expected to sample all three interpretations.

Viewed as a sampling task, this encodes a distribution with three high-probability states, in each of which all assemblies encoding a single, coherent interpretation are active (while all other assemblies are silent). This triple will then inhibit other assemblies due to the WTA structures employed for each sensory modality. We say that the network has found a valid interpretation if one linked assembly triple is active ($> 50\%$ of neurons per assembly fired within the last 10 ms) while all other assemblies remain silent ($< 50\%$ of neurons fired; see Section *Stimulus disambiguation task* in Methods for details). As the recurrent connectivity within each assembly is rather strong, the network tends to lock into one such state, making mixing difficult. However, as generally no interpretation is preferred over the others, the goal of the sampling process is to visit all solutions (with visitation frequencies corresponding to their relative biases) in a reasonable amount of time.

We compared the behavior of the network for oscillating $\alpha$ in the same frequency bands as above ($\alpha \in [0.5, 5]$, i.e., total background rates in [2.5, 25] kHz) with a constant-background

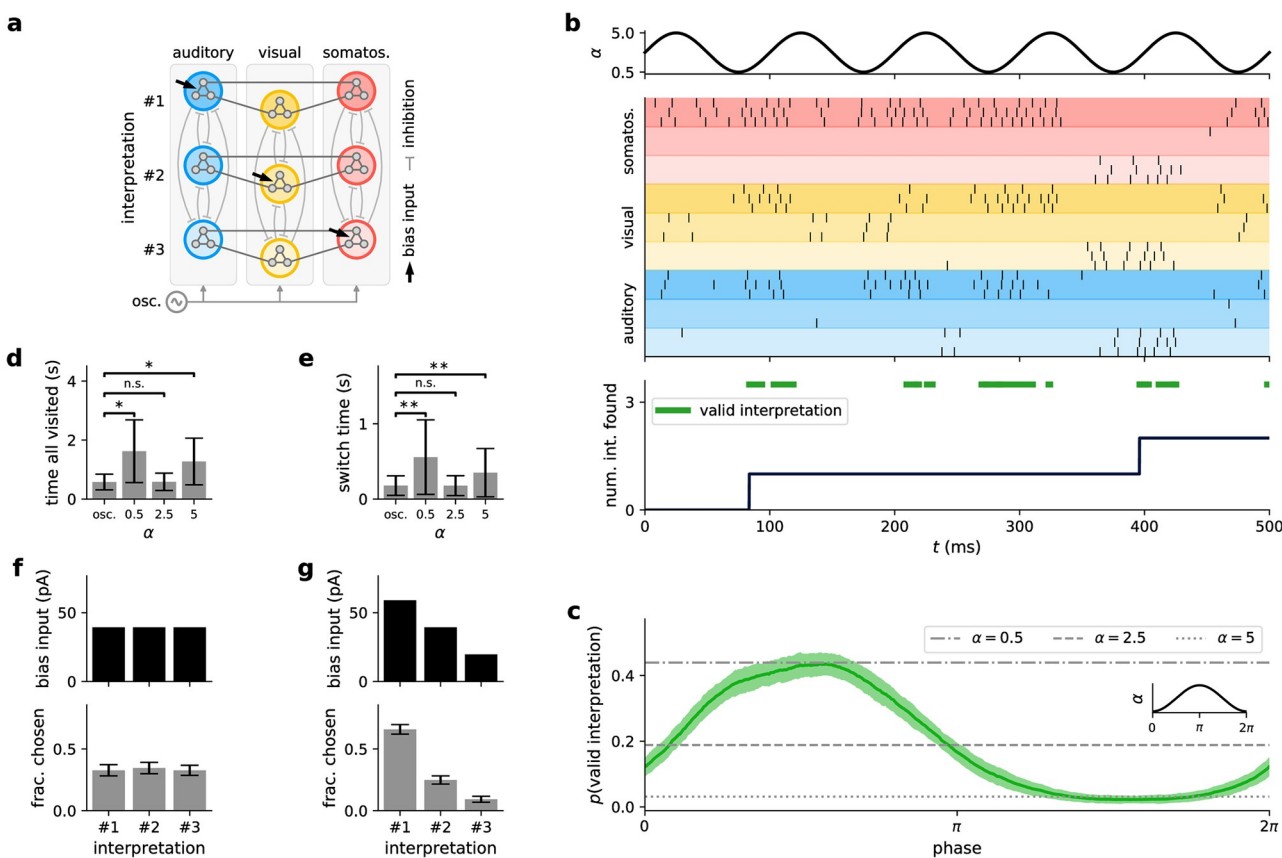

**Fig 6. Background oscillations structure computations into sampling episodes in conductance-based networks. (a)** Setup of stimulus disambiguation task. Assemblies (large colored circles) encode interpretation of stimulus spanning three different sensory modalities (auditory, visual, somatosensory). For each modality, every interpretation is encoded by the activity of a single assembly (different interpretations are represented by different color hues), with lateral inhibition ensuring that one interpretation is chosen. One assembly per sensory modality receives a bias input, resulting in ambiguous input. Simultaneous activity of connected assemblies encoding the same interpretation across all modalities encodes selection of an interpretation (i.e., a solution). All neurons receive oscillatory background input. **(b)** Network activity with oscillating background input. Top: background activity scaling by $\alpha$. Middle: spike raster plot of network activity (color coding of assemblies as in panel a). Bottom: number of solutions (i.e., interpretations) visited so far. Green bars show times in which the network state encodes a solution. **(c)** Probability of network state encoding a solution depending on the phase of the oscillating background input. Gray lines show solution probabilities in networks without oscillating background activity for different activity levels $\alpha \in \{0.5, 2.5, 5\}$. Inset shows background activity phase for reference. **(d)** Mean time until the network has visited all three solutions for oscillating and constant background activity (over $N = 100$ runs for each case). Bars and whiskers show mean and standard deviation, respectively. Significance was calculated with Wilcoxon rank-sum test with $* \doteq p < 10^{-10}$ and n.s. $\doteq p > 0.05$. **(e)** Time between choosing distinct solutions (see Section Methods for details). Plot as in panel d with $** \doteq p < 10^{-100}$. **(f)** For constant bias inputs (top, cf. panel a), solutions are chosen equally often (bottom, means and standard deviations). **(g)** For different bias inputs (top), the probability of choosing the corresponding solution matches the input values (bottom). Plot as in panel f.

scenario. To allow a fair comparison of oscillating to constant background for both low and high background activity, we tuned the synaptic parameters, so the variance of the network firing rates was minimal for different $\alpha$ values (see Section *Stimulus disambiguation task* in Methods for details).

Fig 6b shows network activity over the first 500 ms of a simulation run. The network can jump between attractors and sample different valid interpretations of the ambiguous input. To test whether oscillating background has an advantage over constant input, we performed $N = 100$ simulations lasting 20 s each and calculated the probability for the network state to encode a valid interpretation at any point in time. Fig 6c shows that background oscillations structure sampling-based computations by defining times when good solutions can be read

out from the network. We find that at certain phases, the network with background oscillations has a much higher probability of encoding a valid interpretation than medium or high-level constant background input. Moreover, this probability itself oscillates at the same frequency as the background but with a certain phase lag that depends on the network parameters. In contrast, constant background input produces constant valid-state probabilities, with minimum and maximum values corresponding to those achieved with background oscillations. However, constant background also implies a tradeoff between spending time in valid states and being able to mix between these. For example, networks receiving low background input produce the same valid-state probability as the maximum value achieved with oscillations but tend to converge to one solution and stay there for a long time (see S6 Fig), thus exhibiting much worse mixing behavior.

We quantified mixing by measuring (i) the time it took the network to visit each solution at least once (Fig 6d), and (ii) the time it took on average to move from one solution to another (Fig 6e). On both measures, two regimes achieve comparatively high performance: oscillatory background or constant background at a well-tuned intermediate activity level. However, constant background that is high enough to also facilitate mixing represents a computationally unreliable regime: solutions are found at random points in time and are comparatively ephemeral (cf. S6c and S6d Fig).

In contrast, oscillatory background has a significantly higher probability of producing long-lived valid solutions at well-defined readout phases of the oscillation. Cortical oscillations thus provide explicit temporal structure to sampling-based computations in spiking neural networks. This structure provides good solutions with high probability while inheriting the good mixing properties of high-temperature networks.

For these experiments, we used an equal bias input for each of the three interpretations (Fig 6f, top). Thus, each interpretation should occur equally often. We verified this for oscillating background input (Fig 6f, bottom), where it is indeed the case. In reality, bias input may occur at any level and the resulting frequency of visiting interpretation states changes accordingly. We repeated the analysis ($N = 100$ simulations lasting 20 s) for this case, and found that the visitation levels correspond to the level of bias input (Fig 6g). This shows sampling from a posterior distribution encoding different interpretations of ambiguous input. This highlights a further advantage of background-oscillation-induced tempering in the context of external evidence and sampling from posterior distributions. Since constant background rates can only produce constant temperature, increasing their base level to promote mixing necessarily skews the relative strength of individual attractors by equalizing them (cf. Figs 2, 5g and 5h). Cortical oscillations, on the other hand, preserve the relative dominance of the different modes in the readout phases.

## Constraining sampling models with experimental data

So far, we have shown that sampling networks benefit from temperature oscillations in a variety of parameter regimes. To understand the operating regime of the cortex, it is important to constrain models with experimental data. In this final section, we provide an example of how such links can be established.

We previously discussed how the probability of a valid interpretation changes over the phase of the background input oscillation. One possible way to link the sampling models to experimental data is by considering how the changes within one cycle match recordings from the brain. One study that touches upon this question is [44], which investigated place-cell flickering in rat hippocampus in relation to theta oscillations. In this experiment, the vector of place cell activities was shown to encode the current belief about in which of two possible

chambers the animal was currently situated based on visual cues. The authors computed two prototype activity vectors that represented chamber 1 and chamber 2, respectively. Even in unambiguous situations, the activity vector was not static but occasionally switched to the alternative interpretation for one or a few theta cycles. Typically, the recorded activity was highly indicative of one of the two interpretations of the sensory cues, but over brief periods, activity states were present that were a mixture of the two prototypes. Jezek et al. showed that these mixed interpretations are more likely in the first half of the cycle. These findings are indicative of a sampling-based representation of the animal location in the hippocampus, where one sample is drawn within one theta cycle. In addition, the presence of mixed states indicates a tempering-like sampling procedure where the final sample is formed over a theta cycle. When modeling this data in our sampling framework, the expected ratio of inhibitory to excitatory background conductances $\mathbb{E}[g_{\mathrm{inh}}/g_{\mathrm{exc}}]$ is a free parameter that has a profound effect on the network behavior and the appearance of mixed interpretations. We will see below that this parameter can be constrained by the experimentally observed theta phase of mixed states.

We considered a circuit model with two assemblies encoding correct and incorrect interpretations of the spatial context, see Fig 7a. Each assembly consisted of 20 conductance-based LIF neurons with sparse excitatory connections within the assembly (connection probability 0.1) and lateral inhibition between neurons of different assemblies (see Section *Relating model features to experimental data* in Methods for details). One of the two assemblies received a positive bias current in addition, mimicking strong evidence for its interpretation. The model parameters were chosen such that the flickering was similar to the data shown by [44] at all ratios $\mathbb{E}[g_{\mathrm{inh}}/g_{\mathrm{exc}}]$, i.e., occasional switches to the incorrect interpretation appeared in an unambiguous situation (strong bias current to one of the two assembles), see Fig 7b. This behavior arises from the oscillatory background input in conjunction with the recurrent excitation within and the lateral inhibition between assemblies, leading to a lock-in into one interpretation (i.e., strong activity of one assembly) per cycle. The assembly encoding the correct interpretation is favored because its neurons receive a bias input current, but due to the stochastic nature of the network, the incorrect interpretation is also chosen occasionally. Note that the slightly slower alternating behavior in the data of [44] might arise from additional mechanisms (see Section Discussion). We next systematically varied the model parameters and found that the model behavior was robust to the variations (see S7 Fig).

We next varied the mean background conductance ratio in the range $\mathbb{E}[g_{\mathrm{inh}}/g_{\mathrm{exc}}] \in [0.75, 3]$ by changing the synaptic weights of the inhibitory background input. We found that $\mathbb{E}[g_{\mathrm{inh}}/g_{\mathrm{exc}}]$ shifts the phase between firing rate oscillations of the network and the background oscillations (Fig 7c). For small values of $\mathbb{E}[g_{\mathrm{inh}}/g_{\mathrm{exc}}]$, the background input provides mostly excitation, thus leading to high network activity when the level of background input is high. For large values of $\mathbb{E}[g_{\mathrm{inh}}/g_{\mathrm{exc}}]$, the converse holds: the background input now provides mostly inhibition, therefore, the network activity decreases when the level of background input is increased. Thus, when increasing the ratio of mean inhibitory and excitatory background input, we found that the network changes from a regime where high activity occurs when background input levels are high to a regime where high activity occurs when background input levels are low. The transition between these two regimes occurred around $\mathbb{E}[g_{\mathrm{inh}}/g_{\mathrm{exc}}] = 1.8$ (Fig 7c). Fitting stochastic models for each value of $\mathbb{E}[g_{\mathrm{inh}}/g_{\mathrm{exc}}]$ showed that at this value, the activation functions are aligned at 0.5 for all values of $\alpha$, corresponding to unbiased sampling. This shows that similar to the current-based case, this regime can be achieved by adjusting the balance of excitation and inhibition in the conductance-based case.

We then calculated the probability of mixed interpretations within background input cycles. To match the procedure of [44], we defined the phase in relation to the network activity

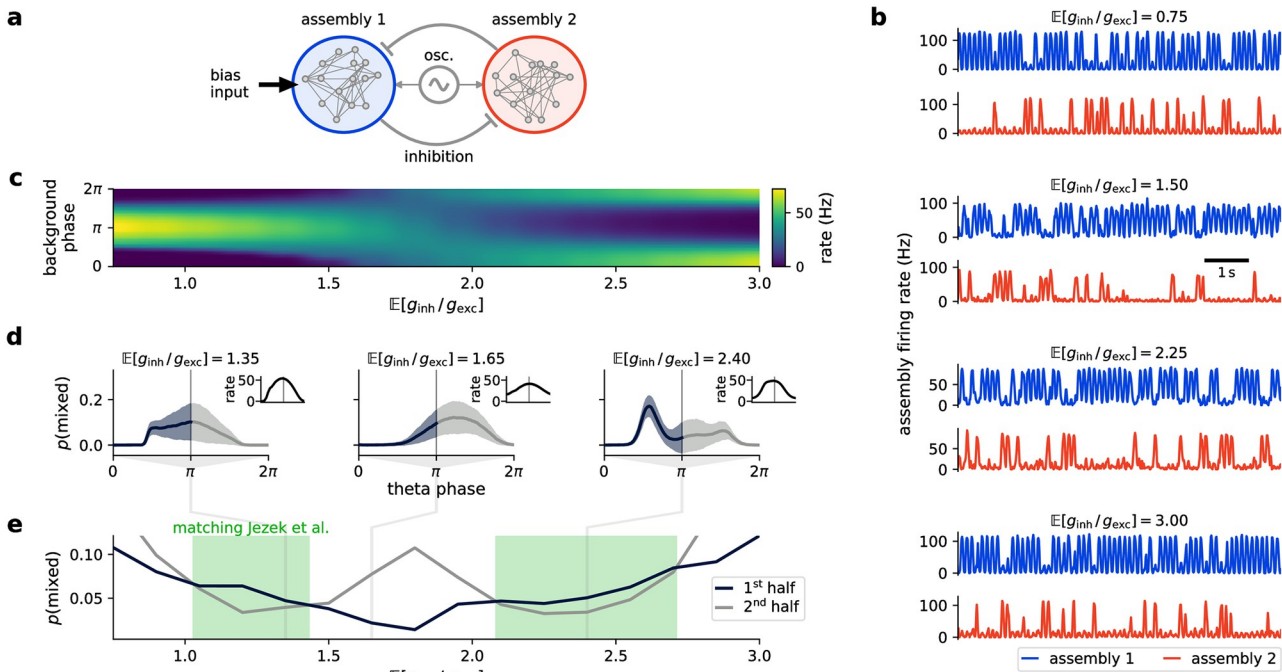

**Fig 7. Relating model features to experimental data. (a)** Simple network model reproducing place-cell flickering behavior. The model consists of two assemblies of spiking neurons with recurrent excitation and lateral inhibition. Assemblies 1 (blue) and 2 (red) encode the correct and incorrect interpretation of the spatial context, respectively. Neurons in assembly 1 receive a bias input current, and all neurons receive oscillatory background input. **(b)** Example model activity (cf. Fig 3c in [44]). Assembly firing rates encoding correct and incorrect interpretations of context are shown in blue and red, respectively. The model behavior holds as the ratio of the mean inhibitory to excitatory background conductance is varied (plot shows $\mathbb{E}[g_{inh}/g_{exc}] \in \{0.75, 1.5, 2.25, 3\}$). **(c)** As $\mathbb{E}[g_{inh}/g_{exc}]$ changes, the minimum firing rate shifts along the background input phase. The background input cycles are defined here as having their onset time (phase zero) when the input rate is at its minimum, i.e., the maximum input occurs at phase $\pi$. **(d)** Probability of mixed interpretations during a cycle. Solid line shows mean, shaded area standard deviation over $N = 100$ network runs of 50 s each. Insets show background input (left) and network firing rate (right, cf. panel b). To match [44], the phase is reordered for every value of $\mathbb{E}[g_{inh}/g_{exc}]$, so that phase zero is aligned with the minimum firing rate in the network (see right inset). **(e)** Mixed state probability in the first and second half of network activity cycle as the mean background conductance ratio is varied. Green shading shows areas in which the mixed probability is larger in the first half of the cycle, matching the results of [44].

recorded in the model (Fig 7c). [44] segregated the network activity into cycles using the recorded firing rate such that the minimum firing rate corresponds to phase zero. We defined the phase at every value of $\mathbb{E}[g_{inh}/g_{exc}]$ accordingly (Fig 7d, insets). We then analyzed the probability of mixed states in the first and second half of the cycle (as [44]) at every value of $\mathbb{E}[g_{inh}/g_{exc}]$ (Fig 7d and 7e). One can observe a strong dependence of the theta-phase of mixed states on this ratio. We found that there were two regions matching the situation described in [44], see Fig 7e. Interestingly, the conductance ratio corresponding to unbiased sampling at $\mathbb{E}[g_{inh}/g_{exc}] = 1.8$ does not fall into either range. This raises the question of whether synapses that mediate the effect of background input on cortical assemblies are optimally tuned towards achieving a balanced regime—i.e., unbiased temperature changes—or whether it is computationally useful for some cortical functions to tune the effect of oscillatory activity towards being explicitly biased.

In summary, we found that a simple circuit model can reproduce theta cycle mediated place cell flickering in the hippocampus. In our model, the inhibition/excitation ratio of the background input determines the theta-phase relation of the network tempering dynamics. This relation is consistent with experimental data from [44] in an unbalanced regime. We emphasize that this experiment only provides a first example of linking sampling models to

experimental data. More data is needed to fully constrain sampling models of the cortex. In particular, experiments establishing more direct links between neuronal properties and sampling model features such as probability distributions or sampling temperatures represent a necessary prerequisite for a quantitative understanding of sampling computations in the brain.

## Discussion

Oscillatory activity is a naturally emerging phenomenon in spiking neuronal networks. As it is well-known that background input increases the variability of neuronal firing, oscillatory background implies oscillatory variability. In the context of ensemble theory, this creates a direct link to the notion of temperature. We have shown that the level of background input determines the sampling temperature in networks of LIF neurons and demonstrated that this effect leads to functional advantages in sampling networks when oscillatory background input is present. This finding holds in the case of current-based synaptic interactions, for which we have presented an analytical treatment of cortical oscillations as tempering, as well as for conductance-based synaptic interactions, for which we have studied a broad range of physiologically relevant parameters in computer simulations. We have furthermore shown that oscillations improve sampling from the distribution represented by the network (i.e., a prior distribution, see Figs 3 and 4) as well as for dealing with uncertainty evoked by input (i.e., posterior distributions, see Fig 6 and S5 Fig). Our results suggest that the ubiquity of oscillations in human and animal brains provides a clear benefit for behaviorally relevant computations, which is elucidated by considering the analogy to simulated tempering.

### Related theoretical work

Our considerations rest on the assumption that for fixed parameters, spiking networks sample from a stationary distribution. This assumption has been shown to hold under only mild constraints for a large class of neuron and network models in [45]. They also showed that in the presence of periodic input, a phase-specific stationary distribution exists, influenced by the network parameters and the properties of the inputs. The existence of such a distribution naturally leads to questions about its specific nature, given specific ensemble dynamics such as those arising in networks of connected LIF neurons and its functional properties for cortical computation. In this work, we have shown that the phase-dependent component is a temperature scaling of a Boltzmann distribution, with periodic background alternating between its exploration and exploitation.

An alternative way of promoting mixing was proposed by [37]. There, short-term synaptic plasticity was shown to weaken local attractors. This mechanism has a similar effect but is different from a change in temperature. Since this form of plasticity only affects active synapses, it only suppresses active local modes rather than flattening the entire distribution. These dynamics ensure that local modes can be abandoned quickly, as synapses can be weakened significantly by only a few spikes, but they come at the cost of changing the sampled distribution. In contrast, cortical oscillations induce a well-defined temporal structure that promotes an undistorted readout. For mathematical tractability, we first considered LIF neurons with current-based synaptic interactions and network structures that are easily amenable to contrastive Hebbian training. We then showed that our results hold for a larger class of biological settings by considering conductance-based synapses and competing neural assemblies. This suggests that the computational role we propose for cortical oscillations is generalizable to a diverse set of cortical structures and their associated functions. Indeed, it has already been observed that oscillations appear to have a similar function throughout the cortex [46].

A similar function of brain rhythms related to slower oscillations was proposed by [21], who suggested on theoretical grounds that during the hippocampal theta cycle, modulation of $GABA_B$ synapses performs a process similar to simulated annealing in a model of population dynamics. Such a mechanism was shown to be advantageous for sequence disambiguation [20]. In this work, we propose that temperature control takes place on the level of individual neurons via input regardless of the synapse type. Thus, the mechanism we propose for incorporating such annealing in neural networks has a much more general scope. [23] showed the benefits of rhythmic changes of neuron excitability in a model of probabilistic memory recall, resulting in a similar kind of annealing as in our model. Our work shows how such a schedule of excitability changes arises in spiking neural networks via background input, thus suggesting an implementation of this mechanism on the cellular level.

One key aspect of previous models is their reliance on excitability modulation of a limited subset of inputs (e.g., recurrent vs. feedforward inputs, [23]). While distinct modulations might arise in biological neurons when inputs target different neuronal compartments, our model shows that this constraint is not necessary to leverage the computational benefits of oscillations as global changes of neuronal input-output behavior suffice. Our model also does not rely on a specific synapse or receptor type, and the proposed mechanism can play out across different oscillatory frequency bands, thus giving our results a very general scope.

The results in this work suggest that oscillations of the background input promote mixing. Previous theoretical work has shown that other sampling methods such as Langevin [47] and Hamiltonian Monte Carlo [24] sampling can also serve this purpose. These studies use rate-based models to sample from continuous-valued probability distributions such as multivariate Gaussians. Our models differ from this approach in two important ways. First, the sampling models based on firing rates [24, 47] require specifically tuned network weights to accomplish rapid sampling. We have shown that oscillating background input can speed up mixing without requiring specifically tuned weights, thus providing our proposed mechanism with a broader scope. Second, our model is based on more complex network state distributions, defined over binary-valued random vectors instead of continuous values. Importantly, these values relate directly to spiking activity. Langevin sampling and Hamiltonian Monte Carlo are not directly applicable to this case. However, it could still be the case that these mechanisms complement each other in the cortex, potentially acting on different timescales.

## Related experimental work and model predictions

Across the entire spectrum of cortical rhythms, individual components of these oscillations are characterized by their frequency and amplitude. We have shown that an effective tempering schedule can be achieved for sinusoidal waves across a wide range of frequencies and amplitudes, roughly corresponding to the range lying between slow and alpha waves [13]. For higher modulatory frequencies, the sampling quality quickly deteriorates as the internal network dynamics cannot react quickly enough to the changes in temperature. However, the soft upper-frequency limit is not fixed and depends on model parameters and the network distribution. In particular, the speed at which the network can change its state depends on the ratio of the dominant time constants of individual neurons and synapses (here on the order of 10 ms) to the duration of a cycle. For faster dynamics, as often observed in vivo (e.g., membrane time constants, [48]; synaptic time constants, [49, 50]; refractory periods, [51, 52]), correspondingly faster oscillations can be accommodated. For example, oscillations in the gamma band could be employed by ensembles with synaptic time constants and refractory times in the order of a few milliseconds, as discussed in recent sampling-based modeling approaches [24, 53]. Thus, this form of tempering can be exploited both for inference in the awake state,

where oscillations are typically fast, and during sleep, for functions such as memory retrieval and consolidation [17–19, 54].

Concerning experimental neuroscience, the suggested computational mechanisms relate to various physiological and psychophysical phenomena, ranging from single-neuron activity to behavior. The tempering in our model modulates the gain of the neuronal transfer functions, similar to the stochastic sampling of a scene through an oscillatory modulation of attentional gain [17, 55], particularly through top-down input [56, 57]. The stochasticity in our model by which stored memories are selectively recalled is mirrored in the randomness of hippocampal replay during sleep that goes beyond the more typical behavior [58], or in free memory recall in humans [59]. The oscillatory recall that supports cognitive computation in our model can also be related to creative thinking [60], to midbrain oscillatory activity during stimulus disambiguation [61], to mind wandering [62] and to local sleep [63].

The oscillation frequency, and thus the rate of temperature change, carries another subtle effect. For slow waves, the effect of a single transition from maximum to minimum temperature is similar to simulated annealing [64]. As the network effectively has more time to relax towards its corresponding thermodynamic equilibrium, it will, at least statistically, tend towards the global minimum energy state. On the other hand, faster oscillations are more akin to tempered transitions [65, 66]. Indeed, the extreme scenario of quenching (extremely rapid cooling) could be implemented by switches between synchronized cortical up and down states [11, 30, 31]. Thus, different oscillatory phenomena in the brain can shift the focus from finding a small set of maximum probability modes to finding a larger range of relevant modes. Similarly, the oscillation amplitude can also control the effective breadth of the exploration space, with larger maximum rates promoting larger jumps between more dissimilar network states.

The benefits of cortical oscillations also extend to other facets of Bayesian inference. For example, when the state distribution is constrained by partial observations, such as in our cue disambiguation task, tempering helps explore the conditional distribution and find multiple ways to solve this pattern completion problem. Similarly, this can help find multiple solutions to a given problem, such as assigning multiple categories to particular input patterns. Importantly, this also highlights the potential benefits of background oscillations during learning (see also [67]), where exploration plays an essential role.

Our results demonstrate that oscillations provide an additional benefit to improved mixing: they serve as a reference for reading out computational results, reducing the amount of data requiring processing, and facilitating the temporal organization of neural computations. Furthermore, they can also serve as a means of input filtering, increasing susceptibility to coherent stimuli [68]. In general, it is well known that information encoding via a background oscillation can be found in the brain, for example, in the hippocampus [69], where place cells convey information by firing earlier or later relative to the theta rhythm. A similar form of coding takes place in our models, as the network distribution changes during each cycle of the background input.

Furthermore, cyclic background input results in the network generating a stream of candidate solutions, with one such state arriving in each cycle. This leads to a form of computing in discrete steps, as computations are structured into episodes defined by background oscillations. A similar type of structured computation has been suggested to take place in monkey and human visual brains during the processing of visual inputs [70]. These experiments showed that shifts in attention were aligned to beta-band oscillations, and every shift took place within a single cycle. In our model, we find similar shifts of the state taking place within each cycle as the temperature decreases.

Temperature changes from oscillations predict that the time course of the network state variability is coupled to the oscillation phase, as we have shown in our model. This suggests

that a similar coupling could be found in sampling-based computations in the brain. [44] have given a hint that this can indeed be the case in hippocampal circuits by showing that ambiguous interpretations of the network input are more likely in the first half of the theta cycle. We have shown how a simple model can reproduce these findings. However, this data is rather coarse, and our results also suggest that a similar behavior can emerge in multiple operating regimes (cf. Fig 7e). Thus, more detailed experimental data are required to constrain sampling models based on background oscillations adequately.

In particular, experimental data could elucidate whether cortical networks are tuned to an unbiased sampling regime. In our model, achieving unbiased sampling requires tuning of neuron parameters and the background oscillation time course. While our analysis based on a simple model of the results of [44] suggested that such a tuning might not be present, a model more closely matching biological networks would help to either corroborate this finding or provide more insight into how such a tuning might be achieved in brain networks. The closer matching could be achieved, for example, by incorporating additional features such as short-term plasticity, neuronal adaptation, and more specific inhibition; see [71] for an example). In general, the balance between excitation and inhibition is of renewed interest in this context, as it connects directly to experimental data. Individual neurons or neuron populations can, for example, use unbalanced rates to implement biases for their associated random variables. Moreover, we expect that different networks tune their background inputs to different balances, depending on which biases are beneficial for their respective tasks.

The experiments of [44] provide evidence for a sampling-based representation of spatial beliefs in the hippocampus, with one sample drawn in each theta cycle. Our model of these results is also based on this basic idea. Another important account of place cell activity states that the activity within different parts of each theta cycle corresponds to different places of the animal within its movement trajectory (e.g., [69, 72]). This view is consistent with a sampling strategy that samples trajectories (temporal sequences) instead of static values, where one trajectory sample is drawn per cycle. While the analysis of trajectory sampling in spiking neural networks is beyond the scope of this work, we note that the general sampling framework can be extended to temporal sequences [45] in which a phase-dependent probability distribution arises from external, phase-dependent input. A model of such a form of sequence sampling could be used for both modeling the phase-dependent activity of neurons encoding previously visited locations [69] as well as for sampling possible future trajectories [72]. For the latter case of sampling diverse sequences within one theta cycle, faster background oscillations (e.g., alpha-band), superimposed on the theta activity, could provide rapid sequential annealing to the individual sequence elements. The extension of our model in these directions is an interesting avenue for future research.

In general, our model relates to simple experimental observations at multiple levels. For example, with respect to the activity of single neurons or small populations, the strength and frequency of cortical oscillations should directly influence the decorrelation of neuronal activity (see also S4 Fig). At a more behavioral level, oscillatory changes in background activity would influence the frequency of perceptual switches. For example, for multi-stable or incomplete images (such as those in S5 Fig), perceptual switches should happen in phases of high activity (i.e., during cortical up-states), as measured, for example, by EEG data. We would thus predict a monotonic relationship between the frequency of switches between up and down states and the frequency of perceptual switches.

In this work, we have used sinusoidal modulations of the background rates. This represents a natural choice, as any other periodic waveform can be described via Fourier synthesis over such elementary waveforms. Particular time courses of the background input would influence and possibly even benefit computations in the network, depending on the circumstances and

nature of the task that needs to be solved. For example, prolonging the low-temperature phase could allow valid samples to be read out over a longer period of time. In contrast, more frequent high-temperature phases would prevent the network from clinging to a possibly wrong belief. The background rates could even take on only two distinct values and alternate between high background activity (resulting in a high temperature, allowing the network to traverse the state space) and low background activity (where the network converges onto a single mode). This provides a link to experiments that study the computational role of cortical on/off states. For example, [73] report that monkeys are more likely to correctly recognize subtle visual cues if they happen during on-states. This aligns with our proposed computational role of cortical background activity, as networks need a stronger background to be able to change their current belief and react to small changes in their input. Note also that these different phases need not be strictly cyclic but might underlie external control, allowing external circuitry to flexibly guide computations in cortical networks according to momentary cognitive demands.

## Applications

Recent years have seen an increasing interest in using spike-based computation on specialized hardware to perform energy-efficient computations [74]. This has spurred efforts to develop models which allow efficient learning and inference with spiking neural networks. Some of these platforms explicitly exploit the stochasticity of their components for computation [75, 76]. By offering a mechanism for modulating neuronal stochasticity, the oscillatory background can enhance computation in stochastic neuromorphic networks, for example, in generative spiking models [8, 77].

Periods of faithful matching between the sampled and target distribution mark the implicit time windows in which computational results can be read out and manifest as constrained intervals of the entire cycle (also see Fig 6c, S1 and S3 Figs). However, it is important to note that the length of these time windows depends on the underlying distribution, the time course of the background modulation, and the time constants in the network. The time window suggests that such oscillations may also improve the performance of networks used for constraint satisfaction problems [78–80]. These are solved by shaping the stationary distribution of the network so that solution states have a high probability. However, it is not clear at any given point in time whether the current state is a solution candidate or a transitional state. In contrast, in an oscillation-driven tempering schedule, it is known that solutions are likely at low-temperature phases.

Overall, the parallels with a variety of empirical phenomena and the advantages for spike-based sampling demonstrated here make neuronal oscillations not only a likely mechanism for supporting stochastic computations in the brain but also a useful tool for fulfilling this same function in biologically inspired neural networks.

## Methods

### Neuron models

**Current-based LIF model.** The membrane potential $u$ of a current-based leaky integrate-and-fire (LIF) neuron evolves according to

$$C_\mathrm{m} \frac{\mathrm{d}u}{\mathrm{d}t} = g_\mathrm{l}(E_\mathrm{l} - u) + I(t) \;, \tag{14}$$

with membrane capacitance $C_\mathrm{m}$, leak potential $E_\mathrm{l}$ and leak conductance $g_\mathrm{l}$. The resulting membrane time constant is $\tau_\mathrm{m} = C_\mathrm{m}/g_\mathrm{l}$. When the membrane voltage $u$ reaches a threshold value $v_\mathrm{th}$ from below, a spike is emitted and the membrane potential is fixed to a reset value $v_\mathrm{reset} \leq v_\mathrm{th}$

for the refractory time $\tau_{\text{ref}}$ (see Table 1). The input current $I(t)$ is a sum of synaptic currents

$$I(t) = I_{\text{rec}}(t) + I_{\text{in}}(t) + I_{\text{bg}}(t) ,$$ (15)

where we distinguish between functional input $I_{\text{rec}}$, synaptic background input $I_{\text{bg}}$ and any other form of bias input $I_{\text{in}}$ (see Fig 1a). Assuming exponential synaptic kernels, the input current obeys

$$\frac{\mathrm{d}I}{\mathrm{d}t} = \frac{I_{\text{in}} - I}{\tau_{\text{s}}} + \sum_j w_j S_j(t) ,$$ (16)

where $w_j$ and $\tau_{\text{s}}$ respectively denote the synaptic weight and time constant. The sum goes over all presynaptic spike sources $j$, including both background and recurrent input, with the corresponding spike trains $S_j(t) = \sum_f \delta(t - t_j^{(f)})$, where $t_j^{(f)}$ denotes the $f^{\text{th}}$ spike time of spike source $j$.

Without loss of generality, we endow each neuron with a single excitatory and a single inhibitory Poisson source characterized by rates $\nu_{\text{exc}}$ and $\nu_{\text{inh}}$ and corresponding connection strengths $w_{\text{exc}}$ and $w_{\text{inh}}$.

The resulting distribution of the free membrane potential $u_{\text{free}}$ (no spiking, $\nu_{\text{th}} \to \infty$) is well described by a Gaussian with moments given by Eqs 1 and 2 (for more details see Section 4.3 in [28]). In general, more background input, originating from either larger weights $w_{\text{exc}}$, $|w_{\text{inh}}|$ or higher frequencies $\nu_{\text{exc}}$, $\nu_{\text{inh}}$, increases the variance. The resulting neuronal response function can be calculated from this distribution using a recursive approach [7]. In the high-conductance state [29], the membrane time constant becomes small, leading to a more symmetric response function, which is well-approximated by a logistic function (Eq 3). In the interpretation of spiking neurons as binary random variables, the neuronal response becomes an expression for the conditional probability of a neuron to be in state "1" given the states of its presynaptic partners $p(z_k = 1 | \mathbf{z}_{\backslash k})$. Neuron parameters are given in Table 1.

**Conductance-based LIF model.** We performed additional experiments with conductance-based models to investigate the behavior in this more biologically realistic case. In this model, $u(t)$ evolves according to

$$C_{\text{m}} \frac{\mathrm{d}u}{\mathrm{d}t} = -g_{\text{l}}(u - E_{\text{l}}) - g_{\text{exc}}(u - E_{\text{exc}}) - g_{\text{inh}}(u - E_{\text{inh}}) ,$$ (17)

where $g_{\text{exc}}(t)$ and $g_{\text{inh}}(t)$ are the excitatory and inhibitory conductances at time $t$, and $E_{\text{exc}}$ and $E_{\text{inh}}$ are the excitatory and inhibitory reversal potentials, respectively. Just like in the current-based case, synaptic kernels are modeled as exponential:

$$\frac{\mathrm{d}g_{\text{exc}}}{\mathrm{d}t} = -\frac{g_{\text{exc}}}{\tau_{\text{exc}}} + \sum_{j \in \text{PRE}_{\text{exc}}} w_j S_j(t) \qquad \frac{\mathrm{d}g_{\text{inh}}}{\mathrm{d}t} = -\frac{g_{\text{inh}}}{\tau_{\text{inh}}} + \sum_{j \in \text{PRE}_{\text{inh}}} w_j S_j(t)$$ (18)

where the sums run over the sets of excitatory and inhibitory presynaptic spike sources, $w_j$ is the quantal synaptic conductance of the synapse with the presynaptic neuron $j$, $\tau_{\text{exc}}$ and $\tau_{\text{inh}}$ are the time constants of excitatory and inhibitory synapses, respectively, and $S_j(t) = \sum_f \delta(t - t_j^{(f)})$ is the spike train of the presynaptic neuron $j$. The spiking mechanism is equivalent to the current-based case. The neuron parameters are given in Table 1.

**Table 1. Neuron parameters.**

| model | $C_{\text{m}}$ (pF) | $g_{\text{l}}$ (nS) | $E_{\text{l}}$ (mV) | $E_{\text{exc}}$ (mV) | $E_{\text{inh}}$ (mV) | $\tau_{\text{exc}}$ (ms) | $\tau_{\text{inh}}$ (ms) | $\nu_{\text{th}}$ (mV) | $\nu_{\text{reset}}$ (mV) | $\tau_{\text{ref}}$ (ms) |
|---|---|---|---|---|---|---|---|---|---|---|
| current-based | 200 | 2000 | −50 | | | 10 | 10 | −50 | −55.1 | 10 |
| conductance-based | 250 | 25 | −65 | 0 | −80 | 2 | 3 | −50 | −65 | 3 |

Unlike in the current-based case, the variance of the free membrane potential has a non-monotonic dependence on background rates $v$, becoming inversely proportional with $v$ for intense background input. This is a consequence of the decreased effective membrane time constant

$$\mathbb{E}[\tau_{\text{eff}}] = \frac{C_{\text{m}}}{g_{\text{l}} + \mathbb{E}[g_{\text{exc}}] + \mathbb{E}[g_{\text{inh}}]} = \frac{C_{\text{m}}}{g_{\text{l}} + w_{\text{exc}}\tau_{\text{exc}}v_{\text{exc}} + w_{\text{inh}}\tau_{\text{inh}}v_{\text{inh}}} \tag{19}$$

which also decreases the amplitude of the spike-induced PSP. The resulting membrane potential distribution is still a Gaussian with moments (Section 4.3 in [28, 40]):

$$\mu_u = \mathbb{E}[u]_{\text{COBA}} = \frac{g_{\text{l}}E_{\text{l}} + \sum_{x \in \{\text{exc,inh}\}} w_x v_x \tau_x E_x}{g_{\text{l}} + \sum_{x \in \{\text{exc,inh}\}} w_x v_x \tau_x} \quad \text{and} \tag{20}$$

$$\sigma_u^2 = \text{Var}[u]_{\text{COBA}} = \sum_{x \in \{\text{exc,inh}\}} \frac{v_x w_x^2 (E_x - \mathbb{E}[u])^2}{2(\mathbb{E}[\tau_{\text{eff}}] + \tau_x)} \left(\frac{\mathbb{E}[\tau_{\text{eff}}]\tau_x}{C_{\text{m}}}\right)^2. \tag{21}$$

The variance depends non-monotonically on the input rates as both $\mathbb{E}[\tau_{\text{eff}}]$ and $\mathbb{E}[u]$ depend on $v_x$.

In the low-input limit ($g_{\text{l}} \gg g_{\text{exc}}, g_{\text{inh}}$), $\tau_{\text{eff}}$ is largely independent of the synaptic conductance, and the generated membrane distribution $p(u)$ behaves similarly to the current-based case. In particular, the variance increases with both increasing input rates $v_{\text{exc}}$ and $v_{\text{inh}}$ and increasing synaptic weights $w_{\text{exc}}$ and $w_{\text{inh}}$ (see Eq 2). In the high-conductance limit, i.e., $g_{\text{l}} \ll g_{\text{exc}}, g_{\text{inh}}$ and thereby $\mathbb{E}[\tau_{\text{eff}}] \to 0$, the variance becomes largely independent of the synaptic strengths and inversely proportional to the input rates $v_{\text{exc}}$ and $v_{\text{inh}}$ (see Section 4.3 in [28]):

$$\lim_{\sum_{x \in \{\text{exc,inh}\}} v_x \to \infty} \text{Var}[u]_{\text{COBA}} = \frac{\sum_{x \in \{\text{exc,inh}\}} w_x^2 v_x \tau_x (E_x - \mathbb{E}[u])^2}{\left(\sum_{x \in \{\text{exc,inh}\}} w_x v_x \tau_x\right)^2} \propto \frac{1}{\sum_x v_x \tau_x}. \tag{22}$$

However, functional synaptic input also has a decreasing effect with increasing background conductance. The result of these two opposing phenomena is that the effect of increasing background rates on the neuronal response function in the conductance-based case matches the one for current-based neurons (see Eqs 3 and 12 and Fig 5e).

## Spike response of sampling neurons

In the experiments underlying Fig 1, we connect a current-based sampling neuron with one excitatory and one inhibitory Poisson source with weights $w_{\text{exc}}$ and $w_{\text{inh}}$, where $w_{\text{exc}} = -w_{\text{inh}}$, and vary the corresponding firing rates $v_{\text{exc}}$ and $v_{\text{inh}}$. Background input parameters are listed in Table 2. We can freely choose the mapping of background rates to the Boltzmann temperature. For simplicity, we chose $T = 1$ in the lower range of physiological values, such that the readout can happen at low points in the oscillation cycle:

$$T = 1 \Leftrightarrow v_{\text{exc}} = v_{\text{inh}} = 2 \text{ kHz} \tag{23}$$

which results in a slope of $\beta = 1.39$ nA$^{-1}$ and an offset of $I_0 = 1.34$ nA (see Fig 1d and 1e). Since shifting the offset implies a change of the neuronal bias, we only have one degree of freedom when changing the temperature $T$. We, again arbitrarily, choose the excitatory rate $v_{\text{exc}}$. The relationship $v_{\text{inh}} = h(v_{\text{exc}})$ is then found by interpolating the measured response functions

**Table 2. Hierarchical network parameters.**

| network | number of neurons | $v_{exc,min}$ (kHz) | $v_{exc,max}$ (kHz) | $v_{inh,min}$ (kHz) | $v_{inh,max}$ (kHz) | $f_{osc}$ (Hz) | $w_{exc}$ (nA) | $w_{inh}$ (nA) |
|---|---|---|---|---|---|---|---|---|
| response function (see Fig 1) | 1 | 0.5–30 | 0.5–30 | 0.4–31 | 0.4–31 | const. | 0.5 | -0.5 |
| entropy (see Fig 2) | 4 | 0.25 | 10 | 0.1 | 10.3 | 1 | 1.0 | -0.5 |
| NORB (see Fig 3) | (3600, 500, 10) | 0.5 | 20 | 0.4 | 20.6 | 1 | 0.5 | -0.5 |
| MNIST (see Fig 4) | (784, 400, 10) | 0.5 | 22 | 0.4 | 22.7 | 2 | 0.5 | -0.5 |

from Fig 1e. In practice, this function can be approximated with the linear fit in Eq 9 with $v_0 =$ −0.13 kHz, $m = 1.04$ and the coefficient of determination $r^2 = 0.999\,98$ (Fig 8a). Choosing inhibitory and exitatory rate combinations along this line, keeps the offset current constant and varies solely the temperature (Fig 8b), which results into response functions with constant inflection point and varying slope (Fig 8c).

The five explicitly marked background configurations shown in Fig 1b–1e are given in Table 3.

### Temperature as a function of background rates

The relationship between our temperature definition and the background rates can be approximated by linking the probability density function of the membrane potential to the derivative of the logistic response function. In the diffusion approximation, the free membrane potential distribution is Gaussian:

$$f(u; \mu_u, \sigma_u) = \frac{1}{\sqrt{2\pi}\sigma_u} \exp\left(-\frac{(u-\mu_u)^2}{2\sigma_u^2}\right) \; . \tag{24}$$

In the high-conductance state, the cumulative distribution function (CDF) has a very similar shape to the (logistic) response function (Eq 3). In particular, they have approximately the same derivative at their inflection point (for details, see [7]). With the parameter transformation $u_{in} = I_{in}/g_l$ and $\beta = \beta_u g_l$, where $\beta_u$ is the slope in the potential domain, the response function reads:

$$v_{out}(u_{in}) = \frac{1}{1 + \exp(-\beta u_{in}/g_l)} \; . \tag{25}$$

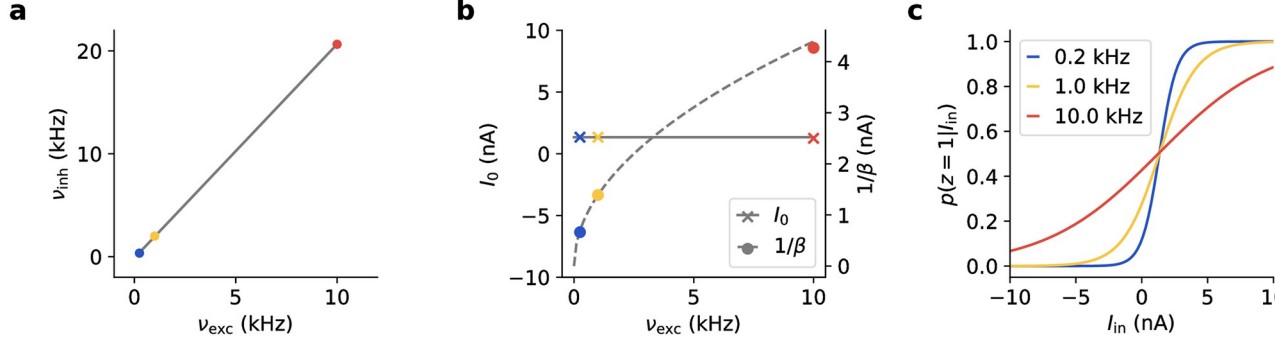

**Fig 8. Changing the temperature of the system.** The linear relationship $v_{inh}(v_{exc})$ in **(a)** can keep the offset of the response function constant (solid line in **(b)**), while only changing the slope of activation functions and thereby the temperature of a network (dashed line line in **(b)**). This relationship also reflects the relative strengths of afferent excitatory and inhibitory weights. **(c)** Three example activation functions with constant offset for different background rates. Colors correspond to those in panel a. Here, we emphasize the binary-state interpretation by plotting $p(z = 1|I_{in}) = v_{out}\tau_{ref}$ (cf. Fig 1c).

**Table 3. Background parameters for the colored response functions and membrane potential distributions in Fig 1.**

| color | $v_{\text{exc}}$ (kHz) | $v_{\text{inh}}$ (kHz) |
|---|---|---|
| blue | 1.000 | 1.000 |
| orange | 2.000 | 2.000 |
| red | 4.000 | 4.000 |
| green | 2.848 | 1.232 |
| purple | 1.232 | 2.848 |

The slope of the CDF at its inflection point is

$$\partial_u \text{F}|_{u=0} = f|_{u=0} = \frac{1}{\sqrt{2\pi}\sigma_u} \ , \tag{26}$$

whereas for the activation function it is

$$\partial_{u_{\text{in}}} v_{\text{out}}|_{u_{\text{in}}=0} = \frac{\beta \exp(-\beta u_{\text{in}}/g_{\text{l}})}{g_{\text{l}}(1 + \exp(-\beta u_{\text{in}}/g_{\text{l}}))^2}\bigg|_{u_{\text{in}}=0} = \frac{\beta}{4g_{\text{l}}} \ . \tag{27}$$

Equating the two creates a direct correspondence between the inverse temperature $\beta$ and the width $\sigma_u$ of the free membrane potential distribution:

$$\beta(\sigma_u) \approx \frac{4g_{\text{l}}}{\sqrt{2\pi}\sigma_u} \ . \tag{28}$$

In our case, with $w_{\text{exc}} = w_{\text{inh}}$, $\tau_{\text{exc}} = \tau_{\text{inh}}$ and $1/k_{\text{B}} = \beta_{\text{ref}}$, plugging in the expression for $\sigma_u$ from Eq 2, the more precise expression for the temperature in Eq 7 is given by

$$T = \frac{\beta_{\text{ref}}}{\beta} = \sqrt{\frac{v_{\text{exc}} + v_{\text{inh}}}{v_{\text{exc,ref}} + v_{\text{inh,ref}}}}, \tag{29}$$

where $v_{\text{exc,ref}}$ and $v_{\text{inh,ref}}$ are the excitatory and inhibitory reference rate corresponding to $T = 1$.

## Entropy of spiking sampling networks

Networks of current-based LIF neurons can sample, to a very good approximation, from binary Boltzmann distributions

$$p(\boldsymbol{z}) = \frac{1}{Z}\exp\left(\frac{-E(\boldsymbol{z})}{k_{\text{B}}T}\right) \tag{30}$$

with energy function

$$E(\boldsymbol{z}) = -\frac{1}{2}\sum_{k,j} W_{kj}z_k z_j - \sum_k B_k z_k \ . \tag{31}$$

where $W_{kj}$ is a symmetric zero-diagonal matrix and $B_k$ a bias vector [7]. The associated neuronal response function represents a conditional state probability and reads

$$p(z_k = 1|\boldsymbol{z}_{\backslash k}) = \frac{1}{1 + \exp(-\sum_j W_{kj}z_j - B_k)} \ . \tag{32}$$

The synaptic strength $w_{kj}$ and input current $I_{\text{in},k}$ in the equivalent LIF network can be related to the Boltzmann parameters $W_{kj}$ and $B_k$ via the slope of the response function $\beta$ (cf. Eq 3):

$$w_{kj} = \frac{W_{kj}}{\beta} \frac{g_{\text{l}}(\tau_{\text{s}} - \tau_{\text{m}})}{\tau_{\text{s}}(1 - \exp(-1)) - \tau_{\text{m}}\left(1 - \exp\left(\frac{\tau_{\text{ref}}}{\tau_{\text{m}}}\right)\right)} \;, \tag{33}$$

$$I_{\text{in},k} = \frac{B_k}{\beta} + I_0 \;. \tag{34}$$

Biases are implemented via a shift of the leak potential $E_{\text{l}}$. The entropy is given by

$$S(p_T) = \sum_{\mathbf{z}} - p_T(\mathbf{z}) \log p_T(\mathbf{z}) \;. \tag{35}$$

Depending on the base of the logarithm, the unit of $S$ is either nats or bits.

**Parameter choice of the Boltzmann distribution.**   For the entropy scaling in Fig 2, we use a 4-neuron network with random weights and biases distributed according to

$$\hat{W}_{kj} \propto \mathcal{N}(0.0, 0.5) \qquad W_{kj} = \frac{\hat{W}_{kj} + \hat{W}_{jk}}{2} \qquad B_k \propto \mathcal{N}(0.0, 0.5) \;, \tag{36}$$

where $\mathcal{N}(\mu, \sigma)$ is the normal distribution with mean $\mu$ and standard deviation $\sigma$. The third and fourth neuron's bias is set to $\pm1$ to ensure one leak-over-threshold and one leak-below-threshold neuron for Fig 2e.

## Image generation examples: NORB and MNIST

The layer sizes of our hierarchical networks are given in Table 2.

**NORB.**   In order to reduce the pixelation in Fig 3a we do not plot the visible state $\mathbf{v} \in \{0, 1\}^{3600}$ directly but instead show the activation probability $p(\mathbf{v})$ that is imprinted by the instantaneous state of the hidden layer:

$$p_{T=1}(\mathbf{v}|\mathbf{h}) = \frac{1}{1 + \exp(-\mathbf{W}_{vh}\mathbf{h} - \mathbf{B}_v)} \;. \tag{37}$$

The temperature schedule of the oscillating background case can be found in Table 2. For the static background input we use the reference configuration (Eq 23) and retrieve samples every $1/f_{\text{osc}} = 1$ s in order to get an equal-time comparison.

**MNIST.**   In Fig 4 we use a similar network structure to the one in Fig 3, with parameters from [37]. Background configurations are varied according to Eq 9 as before and sine parameters are given in Table 2.

**Kullback-Leibler divergence.**   The Kullback-Leibler divergence is a standard measure of the discrepancy between two probability distributions. Intuitively, it measures how many bits are wasted when encoding a distribution $Q$ according to the optimal encoding for distribution $P$. For a discrete probability distribution $P$ with respect to another $Q$, this divergence is defined as:

$$D_{\text{KL}}(P||Q) = \sum_{i} P(i) \log\left(\frac{P(i)}{Q(i)}\right) \;. \tag{38}$$

Note that $Q$ must be strictly positive, whereas $P$ may have states with zero probabilities associated with it.

**Indirect sampling likelihood.** We quantitatively evaluate how well the samples generated by our networks reflect the target distribution by calculating the indirect sampling likelihood (ISL) described in [38]. The ISL measures the similarity between the generated samples and samples from the dataset that were not shown during training (test set). Each test sample $y_j$ and generated sample $x_i$ is a d-dimensional binary vector, whereby each $x_i$ is given by the instantaneous visible layer activity $v \in \{0, 1\}^d$.

For retrieving the ISL, a density model $\mathcal{P}$ is trained on $N$ generated samples, and the likelihood of each test sample under $\mathcal{P}$ is calculated. For $d$-dimensional binary vectors, a non-parametric kernel density estimator is suitable:

$$\mathcal{P}(\boldsymbol{y}) = \frac{1}{N} \sum_{i=1}^{N} \prod_{j=1}^{d} \gamma^{1_{y_j = x_{ij}}} (1 - \gamma)^{1_{y_j \neq x_{ij}}} \; , \tag{39}$$

which is essentially a mixture model representing the $x_i$. The hyperparameter $\gamma \in [0.5, 1)$ determines how much the empirical distribution over $x_i$ is smoothed out (we use $\gamma = 0.95$).

The two exponents denote identity functions that compare an individual test to a generated sample and count the identical and different pixels, respectively. Intuitively, the ISL penalizes each test sample far from any generated sample.

In Fig 4f, we plot $\log \mathcal{P}(\boldsymbol{y})$ averaged over all test samples versus the number of samples. This time course reveals how many main modes of the target distribution are well covered and how fast. Note that the ISL does not necessarily evaluate how diverse the network output is, but rather how well the test set is covered—repetitive samples would yield a high ISL compared to a not very diverse test set.

For orientation, we show the ISL curves for the optimal sampler (OPT) and the product of marginals (POM) (see Fig 4f). The optimal sampler draws randomly, without replacement, from a pool of $10^5$ images that were generated with Adaptive Simulated Tempering (AST) [36], a complex algorithm that is constructed for optimal mixing properties. The POM sampler generates examples by independently sampling each vector component from its respective intensity distribution over the training set. Hence, the marginal probability distribution for each component is preserved, and correlations between components, i.e., the overall structure, are discarded. Note that since these off-class samples overlap significantly with all image classes, they can be associated with higher ISL values than a series of samples from a single mode.

One known drawback of the ISL is that it does not represent an accurate reflection of a human's perceptual judgment of image quality [81]. Therefore, we additionally checked the sampling quality by eye and evaluated the activation probability of the visible layer as shown in Fig 4a and 4b. Based on this, we picked a point on the $f_{\mathrm{osc}} = 0.5$ Hz plane with an intermediate ISL value for display in Fig 4a, 4b, 4e and 4f.

**Mode duration as a measure of mixing speed.** We calculate the mode duration as the average time between two mode switches, where the current mode is defined as the most active label unit, as measured by its probability inferred from the hidden layer activity. The label layer reflects the network's interpretation of its own current visible state $v$ and as such requires the network to be self-consistent. In practice, we did not find significant deviations (see Fig 4a and 4b) from this assumption. Due to computational constraints, we only simulated 1000 s in a single run and averaged over multiple simulations for improved statistics. Note that conventionally, mixing speed is measured by the area under the autocorrelograms of the network neurons' activity, where a smaller area corresponds to faster mixing. For comparison, we also recorded this measure from the inferred spike probabilities of the label neurons, which

confirmed the speed-up in mixing with oscillations (see S4 Fig). However, when classes are discrete, like in the MNIST data set, mode durations are a sufficient and intuitive measure of mixing speed.

## Conductance-based input scenarios

Fig 5a and 5b show the different behaviors of Var[$u$] using the neuron parameters given in Table 1. For the conductance-based models, we investigated four scenarios for conductance-based background input chosen to cover a range of behaviors of $\mathbb{E}[u]$ and Var[$u$] for increasing background input frequencies $v_{exc}$ and $v_{inh}$ (see *Impact of conductance-based synaptic input*). This is accomplished by different ways of changing the rates $v_{exc}$ and $v_{inh}$ with the scaling parameter $\alpha$ (see Eq 11) and different values of the background input weights $w_{exc}$ and $w_{inh}$. All parameters are given in Table 4.

Using Eq 20, we can balance the membrane potential at a target value of $\hat{u} = -55$ mV by choosing $v_{inh,1}$ and $v_{inh,0}$ (see Eq 11 and Table 4) so $v_{inh}$ changes as

$$v_{inh} = v_{exc} \frac{\tau_{exc} w_{exc}(E_{exc} - \hat{u})}{\tau_{inh} w_{inh}(\hat{u} - E_{inh})} + \frac{g_l(E_l - \hat{u})}{\tau_{inh} w_{inh}(\hat{u} - E_{inh})} \quad . \tag{40}$$

## Fitting of stochastic models

To assess the effect of background input on individual neurons' temperature in the conductance-based case, we generated data using realistic inputs to conductance-based LIF neurons and used the fitting method described by [41] to obtain a stochastic model.

The stochastic model is identical to the LIF neurons (i.e., membrane potential generation, refractoriness after spike) except for the deterministic spike generation mechanism, which is replaced by a stochastic spike criterion using an instantaneous firing intensity of

$$\rho(t) = \frac{1}{\Delta t} \exp\left(\frac{u(t) - u_T}{T}\right) \tag{41}$$

where $T$ and $u_T$ are parameters (temperature and soft threshold) obtained from the fitting method. Spikes are drawn from a Poisson process with this instantaneous intensity. In our discrete-time simulations, we calculate the probability of a spike within each simulation time step $\Delta t$, which is

$$\Pr(\text{spike in } [t, t + \Delta t]|u(t)) = 1 - \exp(-\rho(t)\Delta t) \ , \tag{42}$$

and draw spikes accordingly.

To fit this model to data recorded from the simulated conductance-based LIF neurons, we estimated the spiking probability given the membrane potential $u$ using a stimulus consisting of 100 inputs (80% excitatory), each firing according to a Poisson process with $f_{stim}$ = 5 Hz. Each input had a synaptic weight drawn from a uniform distribution in $[0, w_{stim,max}]$ where $w_{stim,max}$ is a maximum conductance value (see Table 4) adjusted for each scenario so the LIF

**Table 4. Background input parameters for the different conductance-based input scenarios.**

| scenario | $v_{exc,1}$ (kHz) | $v_{exc,0}$ (kHz) | $v_{inh,1}$ (kHz) | $v_{inh,0}$ (kHz) | $w_{exc}$ (nS) | $w_{inh}$ (nS) | $w_{stim,max}$ (nS) |
|---|---|---|---|---|---|---|---|
| $\mu_u$ unbalanced | 5 | 0 | 5 | 0 | 0.5 | 0.5 | 30 |
| $\mu_u$ balanced at −55 mV | 5 | 0 | 7.3 | −6.7 | 0.5 | 0.5 | 70 |
| high variance | 5 | 0 | 5 | 0 | 2.5 | 3.75 | 70 |
| low excitation | 1 | 0 | 5 | 0 | 1 | 3.75 | 70 |

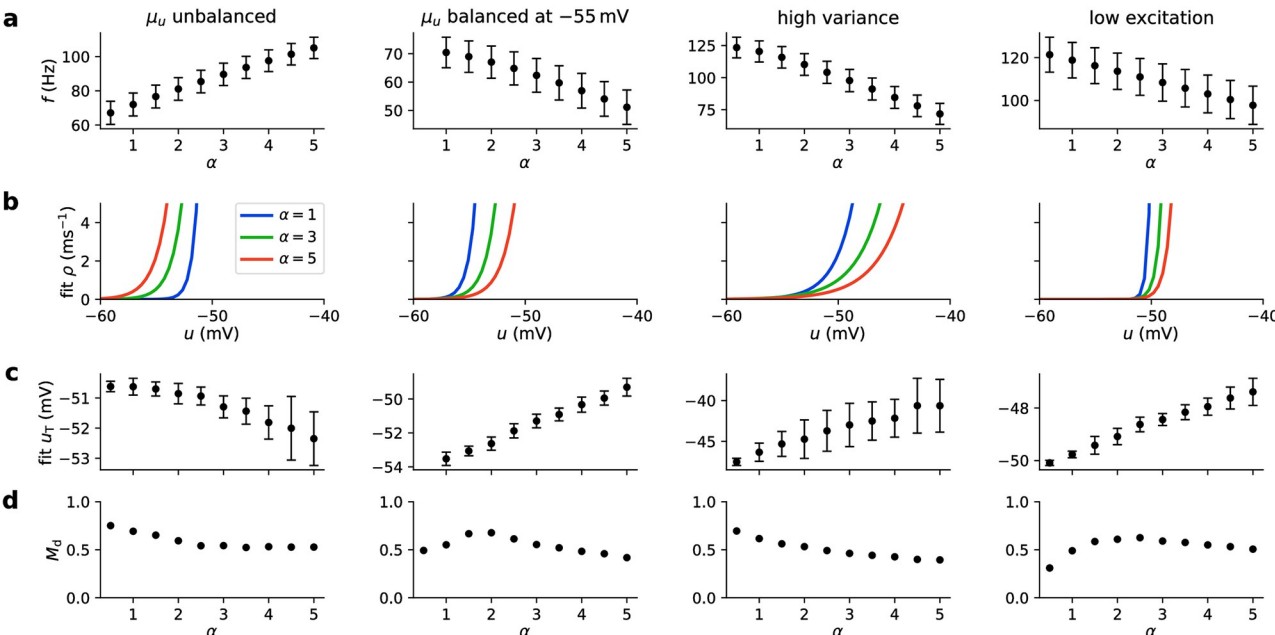

**Fig 9. Details of stochastic model fitting.** (a) Firing rate in response to stimulus used for fitting. Dots denote mean, fliers standard deviation over $N = 50$ independent runs. (b) Result of fitting the stochastic model (see Eq 12) for different values of $\alpha$. The parameters of $\rho$ (including the temperature $T$, Fig 5e) were obtained by averaging the results of $N = 50$ independent fits. (c) Soft threshold values resulting from fitting. (d) Quality of fit as described by $M_d$ criterion (see Methods), measuring match of firing intensities of LIF model and stochastic models resulting from fitting. The fit models reproduce spiking behavior reasonably well ($M_d = 1$ indicates a perfect match, $M_d = 0$ indicates no overlap of PSTHs).

neuron fires at a reasonable rate (i.e., 50 Hz $< f <$ 150 Hz, see Fig 9a). This stimulus was presented to the deterministic LIF models while recording the output spike times. An identical spike train was then presented to a passive version of the stochastic neurons (no firing mechanism, i.e., its membrane potential is the free membrane potential) which was also reset at every spike of the deterministic model. Binned histograms of $u$ of the passive model at all times and at spike times of the original model allow estimating the firing probability $p(\text{spike}|u)$ (Fig 9b; see [41]). To fit the model, we insert Eq 41 into Eq 42 and reformulate the result to get

$$\frac{u - u_{\mathrm{T}}}{T} = \log\left(-\log\left(1 - p(\text{spike}|u)\right)\right), \tag{43}$$

where we perform linear regression on the right-hand side to get values of $T$ and $u_{\mathrm{T}}$. The shape of $p(\text{spike}|u)$ is roughly bell-shaped. We found that the best fits result from using only the values of $p(\text{spike}|u)$ from $u < \arg\max_u p(\text{spike}|u)$ for fitting as the values above the peak show increasing background.

We performed this fitting $N = 50$ times with a simulation durations of 100 s in every run and report the mean and standard deviation of the resulting values of $u_{\mathrm{T}}$ and $T$ (Figs 5e and 9c).

The stochastic models were evaluated by simulating 1000 deterministic and 1000 stochastic versions of the model for 1 s using a new stimulus. From these runs, the time-varying firing intensities $v_{\mathrm{LIF}}(t)$ and $v_{\mathrm{fit}}(t)$ were estimated. A criterion for the quality of fit between the fitted and original models was calculated as

$$M_{\mathrm{d}} = \frac{2 \int v_{\mathrm{LIF}}(t) v_{\mathrm{fit}}(t) \mathrm{d}t}{\int v_{\mathrm{LIF}}^2(t) \mathrm{d}t + \int v_{\mathrm{fit}}^2(t) \mathrm{d}t} \tag{44}$$

for every model. This similarity criterion, which determines how well the firing intensities match, is inspired by [42] (The $M_d$ criterion as stated by [42, Eq. 16] seems to contain an error, therefore, it is slightly adapted here, so its properties match those discussed by Mensi et al., i.e., a value of 1 indicates a perfect match, while a value of 0 indicates no match, e.g., if $v_{fit}(t) \equiv 0$). We found that the stochastic models were generally capable of reproducing the LIF behavior reasonably well (i.e., $M_d > 0.5$ for most models, see Fig 9d). The quality of the fit generally decreases when the variance is high (all models in the high variance scenario, large $\alpha$ in the remaining scenario). We used the results from fitting only for elucidating the temperature effect in networks of LIF neurons with background input, so any errors resulting from imperfect fits did not carry over to the experiments showing the functional advantages of background oscillations (where we again used LIF neurons and spiking background input for our simulations).

## Illustrative sampling task for conductance-based networks

To illustrate the effect of the background input activity on the sampling behavior of an SNN with conductance-based LIF neurons, we used a simple winner-take-all (WTA) network. The model consisted of 4 neurons receiving bias input by injecting currents of amplitudes $I_{in}$ = [40, 60, 80, 40] pA, respectively (Fig 5f). Each neuron additionally received input from an external neuron (Poisson spiking at $f$ = 75 Hz, synaptic conductance $w_{in} = w_{stim,max}$, see above and Table 4). Neurons had inhibitory lateral connections (conductance $w_{inh} = 3w_{in}$).

We ran this network for 100 s. From the spikes of each neuron, we computed network states by setting the state $z_j$ of each neuron $j$ to 1 if the neuron fired within the last 10 ms and otherwise to 0 (see [2]). This allowed us to estimate the fraction of time the network spent in each state (Fig 5g). Note that here, the duration of $z_j = 1$ after a spike does not coincide with the refractory period or the synaptic time constant.

We used only the modes (states in which one neuron was exclusively active, i.e., $z_j = 1$ for some $j = j_0$ and $z_k = 0$ for all $k \neq j_0$) to compute the entropy (Fig 5h).

## Stimulus disambiguation task

We illustrate the advantage of oscillatory background input using a biologically relevant task with several distinct modes which are far apart in the state space, thus making mixing hard. The circuit consisted of 3 winner-take-all (WTA) groups (Fig 6a). Every group contained 3 assemblies, each formed by 3 neurons with strong recurrent connectivity (all neuron pairs bidirectionally connected) and lateral inhibition (bidirectional connections between all neuron pairs that are not part of the same assembly). Between groups, the $n^{th}$ assemblies were bidirectionally linked: in every assembly, 2 (different) neurons were connected to the other 2 assemblies (see Fig 6a). One assembly in each group received additional bias input (Fig 6f and 6g). This results in a stimulus disambiguation task with conflicting input. The synapse parameters are given in Table 5.

We found that using the parameters of each of the scenarios led to complete silence in the network for either low or high background input. To obtain network activity for all values of

Table 5. Connection parameters for conductance-based stimulus disambiguation experiments.

| connection | $g$ (nS) | $E_{syn}$ (mV) | syn. delay (ms) |
|---|---|---|---|
| between assemblies | 17 | 0 | 2 |
| within assemblies | 8.5 | 0 | 2 |
| inhibitory | 17 | −80 | 0.1 |

$\alpha$, we adjusted the parameters of the background input. Starting from the $\mu_u$ unbalanced scenario, we increased the inhibitory background input conductance while keeping the excitatory conductance constant. The variance of the network firing rate (for oscillating background input) decreased until a ratio of $w_{inh}/w_{exc} = 1.35$, after which it increased again. We chose the value at the minimum (i.e., $w_{inh} = 0.675$). This results in network activity also for low and high constant background input, which allows us to compare the effect of background oscillations to constant input without favoring the former. As there are no input units exciting the network and the background input to each neuron is insufficient to evoke network activity, it was necessary to inject a current into each neuron, so the neurons did not remain silent. At $w_{inh} = 0.675$, we used $I_{in} = 350$ pA, which resulted in a mean firing rate of $f \approx 17$ Hz (with oscillating background input).

The circuit defines a sampling problem with 3 modes, i.e., interpretations. Network states were defined to encode the $n^{th}$ interpretation if the $n^{th}$ assembly in each group was simultaneously active while all other assemblies remained inactive. An assembly was regarded as active at every point in time if 50% of its neurons (i.e., at least 2 of the 3 neurons within an assembly) fired within the last 10 ms; otherwise, it was regarded as inactive. This definition allows one to characterize the network state at each time step of the discrete-time simulation as a state either encoding a particular interpretation or none.

Background activity was provided to the network via Poisson sources. Each neuron received independent background input, with rates scaled by $\alpha$ as in previous experiments (as in the $\mu_u$ unbalanced scenario, see the first column in Fig 5c–5e and Table 4). The scaling factor $\alpha$ was sinusoidally modulated over time, with $\alpha(t) \in [0.5, 5]$ and modulation frequency $f_{osc} = 10$ Hz (see Fig 6b top). We compared the results in this case to the results when $\alpha$ was kept constant ($\alpha \in \{0.5, 2.5, 5\}$). S6 Fig shows sample behavior for the different cases, highlighting the different behavioral regimes (e.g., locking into one solution for $\alpha \equiv 0.5$, see *Background oscillations and behaviorally relevant sampling tasks*).

To estimate the probability of the network state encoding a solution, we repeated $N = 100$ simulation runs lasting 20 s each for all four background input setups. For the constant $\alpha$ cases, we report the fraction of network states that encode one of the 3 solutions over all runs. For the oscillatory case, we estimated the fractions of states encoding any interpretation as mean and standard deviation over the 100 runs and plot mean and standard deviation (Fig 6c). Here, we used a bias input of 40 nS for the 3 biased assemblies.

We then tested the mixing behavior in two ways. First, we estimated the time it took to find all solutions by running the network $N = 100$ times for 20 s for each of the four background input setups. Second, we recorded how long it took for the network state to visit each of the 3 solutions at least once in each of these simulations (Fig 6d shows mean and standard deviation). If the simulation time was not enough for the network to visit all solutions, the runs were discarded (this only occurred for uneven input and $\alpha = 0.5$, where about 1/3 of the runs were discarded). We also estimated the time it took to switch between solutions (i.e., the mode duration) over these simulation runs. Switching times were defined as the difference between the time the network state changed to any solution state and the time the network state next changed to a solution state for a different solution (i.e., difference between solution state onset times, Fig 6e shows mean and standard deviation). Significance values were calculated using the Wilcoxon rank-sum test.

Fig 6f shows mean and standard deviation of the fraction of valid states for each interpretation for these $N = 100$ runs. To test whether the interpretations are visited according to the bias input, we set different currents for the three biased assemblies and repeated this analysis (Fig 6g).

### Relating model features to experimental data

To relate the behavior of the conductance-based model to the experimental data from [44], we constructed a circuit consisting of 2 assemblies with recurrent connectivity ($p = 0.1$ for one synapse between each pair of distinct neurons, $w = 2.5$ nS, $E = 0$ mV, synaptic delays randomly chosen from a uniform distribution in [1, 3] ms) and lateral inhibition ($p = 0.5$ for one synapse between each pair of distinct neurons, $w = 5$ nS, $E = -80$ mV, synaptic delay 0.1 ms). One of these assemblies encoded the correct interpretation of the spatial context; neurons in this assembly received a bias input of 10 pA. Background input (oscillating at 8 Hz, i.e., at a medium to high theta frequency) was given to each neuron as in the stimulus disambiguation task. We varied the ratio of inhibitory to excitatory synaptic background input weight in [0.5, 2] as this results in different activity regimes (see above). This leads to $\mathbb{E}[g_{\text{inh}}/g_{\text{exc}}] \in [0.75, 3]$. This model again required current injection due to the stark influence of the background input (S8 Fig shows the resulting firing intensity behaviors of stochastic models fitted to LIF neurons at 3 values of the conductance ratio). The injected current required scaling depending on the conductance ratio (current linearly interpolated from $I_{\text{in}} = 40$ nS at $\mathbb{E}[g_{\text{inh}}/g_{\text{exc}}] = 0.75$ to $I_{\text{in}} = 880$ nS at $\mathbb{E}[g_{\text{inh}}/g_{\text{exc}}] = 3$). This model showed flickering behavior similar to the data shown in [44] along the entire parameter range, see Fig 7b.

Fig 7c shows the firing rate of the neurons in the model over the phase of the background input (mean values over $N = 100$ runs lasting $T = 50$ s). At every conductance ratio, we defined the start of the cycle according to the minimum firing rate of the network (as in [44]). We calculated mixed network states (defined as states in which both assemblies were more than 20% active, similar to [44]) over the phase (Fig 7d shows mean and standard deviation over the 100 runs for 3 conductance ratios). Fig 7e shows how the mean probability of mixed states within the first and second half of the cycle change over the parameter range. We also fitted stochastic models to data at each conductance ratio and found that the response functions intersect at $p = 0.5$ around $\mathbb{E}[g_{\text{inh}}/g_{\text{exc}}] = 1.8$.

### Simulation details

The simulations of sampling experiments with current-based neurons were performed with sbs [82] version 1.8.2 with slight modifications. This framework was executed with PyNN [83] version 0.9.1 and NEST [84] version 2.14.0 with a time resolution of $\Delta t = 0.1$ ms.

The simulations using conductance-based models were performed using Brian2 [85] version 2.4.2 with a time resolution of $\Delta t = 0.05$ ms.

### Supporting information

**S1 Fig. Divergence from target distribution during one oscillation period.** Time course of the Kullback-Leibler divergence to the target distribution together with the time course of the temperature demonstrated at the network in Fig 2. The KL-divergence is high for both high temperatures (red dot, quasi-uniform distribution) and low temperatures (blue dot, quasi-single state distribution), indicating that the distributions at these temperatures differ. The divergence is small at the two crossings of $T = 1$, indicating high fidelity representations. The yellow dot indicates the time of the readout.
(PDF)

**S2 Fig. Layerwise spike activity. (a-c)** Spike activity in label, visible and hidden layer of the NORB network in Fig 3 as a function of the phase of the background oscillation. The mean firing rate per neuron oscillates in all three layers in phase with the background.
(PDF)

**S3 Fig. Probability of the label layer.** Exemplary time course of the inferred activity per label neuron over time (lower plot) and the associated state of the visible layer (top bar) of the MNIST network in Fig 4. Spike probability is high and unique during the low activity phases (around the $T = 1$ readout, vertical lines) and lower and distributed over several labels during the high activity phases. The network is typically in a stable response state for a certain time window around the readout. The length of this time window depends on the depth of the modes. (PDF)

**S4 Fig. Layerwise autocorrelograms indicate improved mixing. (a)** Mean Pearson autocorrelation coefficient calculated from the inferred spike probability of the label layer neurons of the MNIST network in Fig 4—for oscillating background (red) and constant background at $T = 1$ (blue). **(b)** Same as (a), for the visible neurons. Autocorrelation is reduced more quickly for the oscillating setup, leading to a smaller area under the curve, indicating faster mixing. (PDF)

**S5 Fig. Inference task with ambiguous input.** Superposition of the first 5421 images of class 8 **(a)** and class 9 **(b)** of the MNIST training data set. **(c)** Superposition of images in a and b. **(d)** Biases of the network to clamp visible layer to the upper part of the image in c and emulate an ambiguous input. **(e)** Distribution over the inferred labels of the MNIST network from Fig 4 in a 100-cycles run averaged over ten random seeds. The imprinted labels 8 and 9 dominate the distribution—the posterior distribution—illustrating the uncertainty of the input. With oscillating background input, the distribution is more balanced. Thus, oscillations can help in inference tasks. Note that the network simultaneously completes the lower part of the ambiguous input image in the visible layer—shown as the inferred visible layer activity for constant background in **(f)** and oscillating background in **(g)**. (PDF)

**S6 Fig. Background oscillations structure computations into sampling episodes in conductance-based networks: Example network activity. (a)** Sample activity for oscillating background input (see Fig 6b for details). **(b-d)** Sample activity for constant background input with $\alpha \in \{0.5, 2.5, 5\}$. (PDF)

**S7 Fig. Relating model features to experimental data: Robustness of model.** For four values of $\mathbb{E}[g_{\mathrm{inh}}/g_{\mathrm{exc}}]$ (rows, as in Fig 7b), we show activity for systematic variations of the model parameters. In each row, the plot on the left corresponds to the plot in Fig 7b with the base parameters indicated. On the right, each column shows sample activity when one of these parameters (see column title) is varied by multiplying it with 0.8 (top panels within each row) or 1.2 (bottom panels within each row). Variations of the inhibition have the largest impact on the model behavior, with decreased inhibition leading to simultaneous activity of both assemblies in some cases. (PDF)

**S8 Fig. Relating model features to experimental data: Additional information.** Firing intensity behavior of individual neurons determined by fitting stochastic models (as in Fig 9b) drastically changes as the mean background conductance ratio $\mathbb{E}[g_{\mathrm{inh}}/g_{\mathrm{exc}}]$ is increased. (PDF)

## Acknowledgments

We thank Wolfgang Maass for the helpful discussions.

## Author Contributions

**Conceptualization:** Michael G. Müller, Karlheinz Meier, Robert Legenstein, Mihai A. Petrovici.

**Data curation:** Agnes Korcsak-Gorzo, Michael G. Müller, Andreas Baumbach.

**Formal analysis:** Agnes Korcsak-Gorzo, Michael G. Müller, Andreas Baumbach, Robert Legenstein, Mihai A. Petrovici.

**Funding acquisition:** Sacha J. van Albada, Walter Senn, Karlheinz Meier, Robert Legenstein, Mihai A. Petrovici.

**Investigation:** Agnes Korcsak-Gorzo, Michael G. Müller, Andreas Baumbach, Robert Legenstein, Mihai A. Petrovici.

**Methodology:** Agnes Korcsak-Gorzo, Michael G. Müller, Andreas Baumbach, Luziwei Leng, Oliver J. Breitwieser, Robert Legenstein, Mihai A. Petrovici.

**Project administration:** Agnes Korcsak-Gorzo, Michael G. Müller, Andreas Baumbach, Robert Legenstein, Mihai A. Petrovici.

**Resources:** Sacha J. van Albada, Walter Senn, Karlheinz Meier, Robert Legenstein, Mihai A. Petrovici.

**Software:** Agnes Korcsak-Gorzo, Michael G. Müller, Andreas Baumbach, Luziwei Leng, Oliver J. Breitwieser, Mihai A. Petrovici.

**Supervision:** Sacha J. van Albada, Karlheinz Meier, Robert Legenstein, Mihai A. Petrovici.

**Validation:** Agnes Korcsak-Gorzo, Michael G. Müller, Andreas Baumbach, Luziwei Leng, Oliver J. Breitwieser, Robert Legenstein, Mihai A. Petrovici.

**Visualization:** Agnes Korcsak-Gorzo, Michael G. Müller, Andreas Baumbach, Robert Legenstein, Mihai A. Petrovici.

**Writing – original draft:** Agnes Korcsak-Gorzo, Michael G. Müller, Andreas Baumbach, Robert Legenstein, Mihai A. Petrovici.

**Writing – review & editing:** Agnes Korcsak-Gorzo, Michael G. Müller, Andreas Baumbach, Sacha J. van Albada, Walter Senn, Robert Legenstein, Mihai A. Petrovici.

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
