## [Decision Letter · Decision Letter 0]

15 Jun 2021

Dear Ms Korcsak-Gorzo,

Thank you very much for submitting your manuscript "Cortical oscillations support sampling-based computations in spiking neural networks" for consideration at PLOS Computational Biology.

As with all papers reviewed by the journal, your manuscript was reviewed by members of the editorial board and by several independent reviewers. In light of the reviews (below this email), we would like to invite the resubmission of a significantly-revised version that takes into account the reviewers' comments.

In particular, both reviewer 2 & 3 highlighted (1) problems with clarity in some sections of the manuscript, and (2) a lack of discussion/consideration of prior work and how this method using oscillations relates to and/or improves on other techniques for sampling from a distribution with a neural network.

We cannot make any decision about publication until we have seen the revised manuscript and your response to the reviewers' comments. Your revised manuscript is also likely to be sent to reviewers for further evaluation.

Sincerely,

Blake A Richards

Associate Editor

PLOS Computational Biology

Daniele Marinazzo

Deputy Editor

PLOS Computational Biology

In particular, both reviewer 2 & 3 highlighted problems with clarity in some sections and a lack of discussion/consideration of prior work and how this method using oscillations relates to and/or improves on other techniques for sampling from a distribution with a neural network.

Reviewer's Responses to Questions

**Comments to the Authors:**

Reviewer #1: The authors consider a sampling-based model of neural activity in which binary (spiking) neurons sample a probability distribution over latent variables conditioned on data. They show that the level of background activity changes the effective temperature in such networks, so oscillating background activity gives rise to a tempering-like setup which encourages mixing between modes.

The paper is convincing, comprehensive and well written. I have no hesitation in recommending acceptance (pending very minor revisions).

Big-picture comments (these are suggestions; I would be happy to recommend acceptance without them being addressed):

The manuscript is long with 9 Figures in the main text. The paper might ultimately have more impact if it were streamlined (with Figures merged or pushed to supplementary).

The section on matching experimental data (Fig. 9) is relatively short. Adding more here would be great (but I am well aware this may not be possible).

Minor comments:

"we unify two individually well-studied, but previously unlinked aspects of cortical dynamics under a common normative framework: probabilistic inference and cortical oscillations."

Given Aitchison and Lengyel (2016), Echeveste et al. (2021) and Savin et al. (2014), the claim that these are "previously unlinked" is too strong. But these papers are not novelty-destroying as e.g. they use rates rather than spikes.

Savin (2014) would seem to be the most relevant prior art, and it is currently a bit buried in the "Related work". It should be disucssed earlier in the Introduction and more extensively.

Eq. 1+2 appear to ignore correlations in the presynaptic inputs. But I buy the overall logic.

"Upon introducing a firing threshold, some portion of the free membrane potential will lie above it"

some portion of the membrane potential *probability density* lies above the threshold.

The figures are occasionally described in a cursory fashion, e.g. "Overall, the best performance was achieved in the slow-wave regime (Fig. 5c-g)".

Could do with a bit more clarity about exactly what equations are actually being run for the simulations (i.e. was it really an LIF in the sims, or was there a Boltzmann Machine with a variable temperature?). If nothing else, it is necessary to state the equations for the LIF, and tell us specifically how the background activity was implemented in the main text.

The experiments for the conductance based network appears considerably more simplistic than those for the current based network. Can the authors comment on the difficulties?

The use of insets in plots should be minimized as it is very difficult to e.g. write on axis labels.

More careful referencing to the relevant part of Methods would be appreciated.

Reviewer #2: Summary

This manuscript presents the intriguing and promising proposal that cortical oscillations may serve the functional role of speeding up computational dynamics, especially the transitions between different modes in multimodal distributions. The paper shows how excitatory and inhibitory background firing rates act together to determine slope and operating point of the response functions of current-based and conductance-based LIF neurons. It interprets the slope of the response function as a computational temperature on the distribution that the cortical states are sampled from. It then implements the idea of rhythmic changes in the background activity in several spiking network models and illustrates the effect visually and using multiple statistics.

While the idea is novel and interesting, I have several major concerns about both implementation and exposition.

1. As far as I know, cortical oscillations are commonly believed to reflect changes in collective mean activity of neurons, not the slope of the input-output function of individual neurons. However, in the authors’ proposal, the mean is kept constant so it’s not clear to me that classical measurements of oscillations would be able to measure them in the authors’ model. Is this correct? Furthermore, recent work (Engels et al. Science 2016) has found that high activity states were associated with high behavioral performance which appears to be in contradiction with the authors’ idea that high (background) rates are associated with a high temperature?

2. How exactly does Eq 4 establish the relationship between ensemble temp and background firing rate? One is the slope of the input-output function, the other the scaling of the joint probability function over all states z. Please elaborate.

3. Currently the computational benefits are presented in the context of model networks sampling from what might best be described as a prior distribution, as opposed to a posterior that’s inferring the correct label for an input image (e.g. for the first two models). The problem with that approach is that the time to transition from one state to another is directly confounded with how close to factorial the distribution is. As far as I can tell, the authors currently have not quantitatively demonstrated that the oscillations do anything actually useful. The hypothesized benefit e.g. during inference, remains a conjecture. A better and more direct test would be to measure how long it takes for one of their models to infer the correct category for an observed image starting at a random state. That time could be compared with and without the proposed oscillations on the temperature verifying that sampling from the wrong distribution during the phases with high temperature is compensated by the higher speeds at which the correct category is reached/inferred.

4. The section comparing their proposal to neurophysiological data is incomprehensible to me. The model remains completely unclear even with the information in the Methods, with no intuitions provided for why the model behaves the way it does. It is also unclear which of the modeling findings are robust to the model details and to what degree the discrepancy between model and data presents a challenge for their proposal in general, or just this particular model. I personally would probably omit this section - the key proposal by authors about the role of oscillations is sufficiently abstract that reliably testing it using empirical data requires multiple uncertain links that might go beyond the scope of this paper.

Exposition:

- The manuscript presents several models without clearly describing any one of them. I think it would help a lot to focus on one model, e.g. the one based on the MNIST dataset, describe the model in detail, as well as clearly demonstrating the effect and benefits of oscillations. The same model could then be implemented using conductance-based LIF neurons without much needed extra explanations.

- The current manuscript assumes too much prior knowledge. When building on prior work it would really help to briefly summarize the key result that is being incorporated into the present work.

- When describing the modeling results, it’s often helpful to move from a individual examples to summary statistics to how summary statistics depend on parameters, not the other way around as currently, e.g. in Fig. 5.

Minor:

Fig 5f: blue distribution impossible to discern: 50% of mass at 2, and 50% at 998?

Fig 5g: does this suggest that the tempered sampler is worse than the factorized one up to 20 cycles? Isn’t that a problem?

How does the background noise affect spiking statistics?

Fig 6: meaning of columns unclear

How does alpha relate to beta relate to temperature?

It would help to more clearly motivate the conductance-based simulations, and how exactly they differ. Currently, applied to different models it is very hard to understand the relevant differences between the two implementations and what they mean with respect to the central claim of the paper.

Consider using autocorrelation to quantify mixing time

Some of the results (e.g. in Fig 5) are presented in terms of cycles, but the results must depend on the frequency of the tempering relative to the biophysical time constants in their spiking model, right? Please make that explicit.

Fig 5: What is DKL(I)? Shouldn’t there be two distributions in the argument?

P.10: Typo “Although νexc = νinh in this case (Fig. 6a, first column), ..’ should be Fig.6c’

What is the motivation for Eqn.10?

Reviewer #3: The authors present a theory on the way recurrent neural networks can perform stochastic computations. In particular, the authors argue that approximate Bayesian computations can be efficiently performed if oscillations are present. The key to their argument is that in order to perform efficient inference sequential samples need to be obtained from a probability distribution (notably, the posterior distribution) and a faithful representation of such a distribution requires fast mixing, i.e. that all possible corners of the state space with finite probability can be visited without being trapped in a particular local mode of the distribution. The starting point of the paper is that a background network of neurons imposes a dynamics on a set of ‘signal neurons’ that stochastically explores the activity space, thus implicitly defining a ‘distribution’ of activities. The authors then provide an elegant argument that under some circumstances the entropy of this distribution can be simply and systematically decreased and thus achieve fast mixing. A minor note here concerns readability: while the argument can be understood from the text, the flow of the text is not linear and bits and pieces of the line of thoughts need to be gathered by going back and forth in the text. The text would benefit from a clear cartoon that walk the reader through the individual steps.

The theory behind controlling the width of the activity distribution provides some interesting insights into network dynamics. However, several critical questions regarding the role of such a process in approximate inference remained elusive.

First, the authors start with the argument that the nervous system needs to be able to represent uncertainty. Representing uncertainty in terms of the variability of neural activity has gained support in recent years. Here, uncertainty affects the entropy of the distribution therefore it would be critical to see if the variability expressed by the network can be related to uncertainty. Critically, increased uncertainty has similar effects to the ‘annealing’ process proposed here. It would therefore be necessary to demonstrate that the two components of the algorithm (representing uncertainty by neural variability and performing annealing) can be achieved using the same substrate. In general, a more principled link between the distribution defined by the recurrent connections and the posterior distribution would be important.

Second, the main focus of the paper is the speeding up of the faithful representation of a probability distribution through sampling. Annealing is indeed a well-established method in statistics to achieve this goal. The proposed format, however, poses a number of problems that remain unaddressed in the paper. First, oscillatory annealing leaves only a narrow window left to read out the true distribution. That is, considering the only discussed frequency range, theta activity, a fraction of the ~100-ms cycle open for sampling the represented distribution. While the paper provides insights into how the oscillation can achieve mixing between modes but the representation of the uncertainty is left open. In a population of neurons a single sample could be obtained in a much shorter time frame than the cycle of theta activity. The cycle of theta activity corresponds to the perceptual time scale, therefore pruning samples by using the oscillation can be detrimental to perception. In sum, it would be important that the representation of uncertainty is not hurt by introducing annealing.

Several studies have previously addressed the question of efficient sampling in neural networks. These rely on techniques that are widespread in machine learning, e.g. Hamiltonian Monte Carlo is core to solutions in probabilistic programming. Unfortunately, the current paper does not build and does not critically review these alternatives. The most direct comparison directly addresses decorrelation of samples in a neural network using balanced neural networks (Guillaume Hennequin, Laurence Aitchison, Mate Lengyel, Fast Sampling-Based Inference in Balanced Neuronal Networks, Advances in Neural Information Processing Systems 27 (NIPS 2014)). This paper has clear parallels in the network structure, objectives, and assumptions with the current paper. The other paper, which is cited but not contrasted with the current proposal is ‘The Hamiltonian Brain: Efficient Probabilistic Inference with Excitatory-Inhibitory Neural Circuit Dynamics’. This paper uses Hamiltonian Monte Carlo a sampling method that excels in helping mixing. Here the role of inhibitory neurons and oscillations is complementary to the current proposal. It would be a natural testbed to contrast the effectiveness of HMC with tempering in overcoming sampling a multimodal distribution.

Minor:

The narrative of the results would benefit from cleaning. Linearity of the argument is hurt a number of times and motivations of the steps are not sufficiently clearly spelled out. Similarly, the introduction section could benefit from a clarification (e.g. sampling at the end of the first paragraph is introduced without much background given, and the paper immediately navigates to ‘mixing’ a term that is quite complex and requires more explanation).

The sentence on p 5 ‘Thus, oscillatory background activity can be interpreted as tempering, a periodic cycle of heating and cooling, with hot phases for mixing and cold phases for reading out the most relevant samples of the correct distribution.’ is little motivated: there is a leap in the argument that is hard to follow, even though the insight is later demonstrated.

‘In doing so, we unify two individually well-studied, but previously unlinked aspects of cortical dynamics under a common normative framework: probabilistic inference and cortical oscillations’ — please specify, as note above this claim is not true

The penultimate paragraph of the introduction ventures towards topics that are not addressed in the results but provide a perspective to the current work, I suggest moving those to the discussion.

Sampling the troughs of oscillation is proposed to obtain decorrelated samples the target distribution. It would be insightful to see an analysis on the width of the time windows that correspond to a faithful representation of the target distribution.

The motivation and details of particular networks used during the results is little motivated it is very hard to keep pace with the manuscript at places where these models are introduced. Also, the link between the text body and the figures is often weak and interpreting figures is not straightforward (a prime example is Fig 3a). Similarly, on Fig 4 it is hard to understand why the particular distribution is relevant and why are there multiple modes? Further, in section 2.4 @ Eq. 9 the motivation for the current form is not clearly provided although reading through the section will yield an understanding. Same for the network architecture in 2.5: very brief descriptions are provided and choices are not sufficiently justified.

Fig 6a,b: a clearer reference to the panels would be useful.

No direct consequences of the proposal has been addressed in the paper that could contrast it with empirical data. For instance, the lower temperature samples would introduce slower decorrelation of spikes, while higher temperature would introduce faster decorrelation.

The empirical data that is presented refers to the ambiguity of place cell representations at different phases of theta. An influential account of place cell activity that has strong ties to functional models claims that place cell activity at different phases of theta oscillations correspond to places at different parts of the animal’s movement trajectory (Brad E. Pfeiffer & David J. Foster, Hippocampal place-cell sequences depict future paths to remembered goals, Nature 497:74–79 (2013)). Since this is the one experimental outlook the results section provides, it would be useful for the reader to contrast the explanatory power of the competing theories.

**Have the authors made all data and (if applicable) computational code underlying the findings in their manuscript fully available?**

Reviewer #1: **No: **No evidence of code availability in the manuscript. For data availability, they left "All XXX files are available from the XXX database (accession number(s) XXX, XXX.)." in the form.

Reviewer #2: None

Reviewer #3: Yes

PLOS authors have the option to publish the peer review history of their article (what does this mean?). If published, this will include your full peer review and any attached files.

Reviewer #1: No

Reviewer #2: No

Reviewer #3: No
---

## [Decision Letter · Decision Letter 1]

8 Nov 2021

Dear Ms Korcsak-Gorzo,

Thank you very much for submitting your manuscript "Cortical oscillations support sampling-based computations in spiking neural networks" for consideration at PLOS Computational Biology. As with all papers reviewed by the journal, your manuscript was reviewed by members of the editorial board and by several independent reviewers. The reviewers appreciated the attention to an important topic. Based on the reviews, we are likely to accept this manuscript for publication, providing that you modify the manuscript according to the review recommendations.

Specifically, please attend to Reviewer 3's remaining concerns. They can all be dealt with be adding to the discussion and/or text in the results.

Sincerely,

Blake A Richards

Associate Editor

PLOS Computational Biology

Daniele Marinazzo

Deputy Editor

PLOS Computational Biology

[LINK]

Please attend to Reviewer 3's remaining concerns when submitting the final version of the paper. They can all be dealt with be adding to the discussion and/or text in the results.

Reviewer's Responses to Questions

**Comments to the Authors:**

Reviewer #1: The authors' response is complete. I have no further comments and recommend acceptance.

Reviewer #2: The authors have adequately addressed my comments and concerns. The accessibility of the manuscript has also substantially improved.

Reviewer #3: I thank the authors for the detailed answers to the issues I raised in my review.

I believe that the readability has improved considerably in the current version of the manuscript.

I have three remaining concerns.

At issue #32 the authors refer to a newly added figure. It is indeed true that multiple interpretations to the same stimulus greatly contribute to uncertainty and this aspect is covered in this analysis. What my comment was referring to is the issue of another form of uncertainty that plagues everyday inference: when a single-mode posterior becomes wider as a result of poorer data (contrast, limited observation, occlusion, etc). This form of uncertainty results in a similar form of widening of the posterior as annealing does. I believe that it is crucial property of sampling that such changes in the posterior can be reflected. The current phrasing of the manuscript suggests that the proposed method offers a full-feldged solution for sampling the posterior but the actual focus is much narrower since the above widening is not covered at all. I believe that covering the concept of such posterior widening would be essential for the readers to have the scope and limitations of the paper.

At issue #33 the authors propose that up/down states can provide additional opportunity for an annealing-like behavior. I find this proposal intriguing, especially because this phenomenon has a wide literature, including papers that feature intracellular recordings (e.g. (Tan, A. Y. Y., Chen, Y., Scholl, B., Seidemann, E., & Priebe, N. J. (2014). Sensory stimulation shifts visual cortex from synchronous to asynchronous states. Nature, 509(7499), 226–229. http://doi.org/10.1038/nature13159), that can provide the necessary means to test the theory. I find it thought provoking that the paper provides links to a number of phenomena but this links remains at the speculative side despite available data.

At issue #48: I am not sure I understand the argument the authors provide. According to the proposed role of oscillations, a neuron population is sampling the same distribution at different phases of the oscillation but the ‘temperature’ of this distribution varies with the phase of the oscillation. The work on the Foster lab (also Loren Frank’s and David Redish’s labs) points out that at different phases of the theta oscillation different portions of the trajectory are sampled that correspond to past present and future locations of the animal. It is hard reconcile this view with the idea that the same distribution is sampled at different phases. Since hippocampal theta is the only point where the paper ventures into actual comparison with experimental data, I believe that clarifying this issue is important.

**Have the authors made all data and (if applicable) computational code underlying the findings in their manuscript fully available?**

Reviewer #1: None

Reviewer #2: **No: **Please provide a link to code that reproduces all non-conceptual figures.

Reviewer #3: Yes

PLOS authors have the option to publish the peer review history of their article (what does this mean?). If published, this will include your full peer review and any attached files.

Reviewer #1: **Yes: **Laurence Aitchison

Reviewer #2: No

Reviewer #3: No

Figure Files:

Data Requirements:

Reproducibility:

References:

---

## [Editor Report · Decision Letter 2]

14 Dec 2021

Dear Ms Korcsak-Gorzo,

We are pleased to inform you that your manuscript 'Cortical oscillations support sampling-based computations in spiking neural networks' has been provisionally accepted for publication in PLOS Computational Biology.

Best regards,

Blake A Richards

Associate Editor

PLOS Computational Biology

Daniele Marinazzo

Deputy Editor

PLOS Computational Biology

---

## [Editor Report · Acceptance letter]

21 Feb 2022

PCOMPBIOL-D-21-00596R2 

Cortical oscillations support sampling-based computations in spiking neural networks

Dear Dr Korcsak-Gorzo,

I am pleased to inform you that your manuscript has been formally accepted for publication in PLOS Computational Biology. Your manuscript is now with our production department and you will be notified of the publication date in due course.

With kind regards,

Zsofia Freund
